# Diet and Nutritional Factors in Male (In)fertility—Underestimated Factors

**DOI:** 10.3390/jcm9051400

**Published:** 2020-05-09

**Authors:** Kinga Skoracka, Piotr Eder, Liliana Łykowska-Szuber, Agnieszka Dobrowolska, Iwona Krela-Kaźmierczak

**Affiliations:** Department of Gastroenterology, Dietetics and Internal Diseases, Poznan University of Medical Sciences, Heliodor Święcicki Hospital, 60-355 Poznań, Poland; kingskoracka@gmail.com (K.S.); piotr.eder@op.pl (P.E.); lszuber@wp.pl (L.Ł.-S.); agdob@ump.edu.pl (A.D.)

**Keywords:** semen quality, male infertility, nutritional model, diet, anti-oxidants

## Abstract

In up to 50% of cases, infertility issues stem solely from the male. According to some data, the quality of human semen has deteriorated by 50%–60% over the last 40 years. A high-fat diet and obesity, resulting from an unhealthy lifestyle, affects the structure of spermatozoa, but also the development of offspring and their health in later stages of life. In obese individuals, disorders on the hypothalamic-pituitary-gonadal axis are observed, as well as elevated oestrogen levels with a simultaneous decrease in testosterone, luteinizing hormone (LH), and follicle-stimulating hormone (FSH) levels. Healthy dietary models clearly correlate with better sperm quality and a smaller risk of abnormalities in parameters such as sperm count, sperm concentration and motility, and lower sperm DNA fragmentation. Apart from mineral components such as zinc and selenium, the role of omega-3 fatty acids and antioxidant vitamins should be emphasized, since their action will be primarily based on the minimization of oxidative stress and the inflammation process. Additionally, the incorporation of carnitine supplements and coenzyme Q10 in therapeutic interventions also seems promising. Therefore, it is advisable to have a varied and balanced diet based on vegetables and fruit, fish and seafood, nuts, seeds, whole-grain products, poultry, and low-fat dairy products.

## 1. Introduction

Infertility, i.e., the inability to get pregnant despite regular, minimum yearly sexual intercourse without using any contraceptives, affects an increasing proportion of society [1,2,3,4,5].

It is estimated that worldwide, as much as 15% of couples, i.e., about 70 million couples of reproductive age, experience problems with getting pregnant, with approximately half of the cases related to male infertility [2,4,6]. It is reported that an estimated 35% of infertility cases involve only women, 20% both women and men, 30% involve problems only on the part of the man, and 15% of infertility cases remain unexplained [1].

There are several characteristic features of male infertility, such as oligospermia, i.e., low sperm concentration in semen; asthenozoospermia, i.e., an absolute lack of motility or a decreased motility of spermatozoa; and teratozoospermia, i.e., an insufficient number of spermatozoa of normal structure [7]. Leaver points out that these disorders constitute over 90% of male infertility causes [1]. According to an extensive meta-analysis covering 185 studies, including over 40,000 men from developed countries, the number of spermatozoa, i.e., the main factor determining the quality of semen, decreased by 50%–60% over the period 1973–2011 [8]. According to research carried out in Poland on a group of 169 young, healthy men with unknown fertility status from the Lower Silesia region, the average and median of seven parameters determining sperm quality were within the limits of the WHO standards in 2010. However, sperm viability was close to the lower range of the norm, whereas the average percentage of abnormal sperm structure was as high as 85%. Nearly 9% of the studied cases had one, two, or three parameters outside the limits of the standard [9]. 

Environmental factors that significantly affect male fertility comprise smoking cigarettes and cannabis, anabolic steroid use, excessive alcohol consumption, emotional stress, excessive exposure to high temperatures, age, tight clothing, environmental pollution, sedentary lifestyle, exposure to pesticides and toxins, radiofrequency electromagnetic radiation, as well as cytotoxic drugs, cadmium, and lead [1,6,10,11,12,13,14,15,16,17,18]. It is vital to bear in mind that some factors such as age, environmental pollution, or radiation cannot be avoided [12,16]. However, some research studies suggest that the use of antioxidants, such as resveratrol, may constitute a therapeutic alternative [17].

Furthermore, recent research data point to the fact that diet is also directly associated with semen quality and that overall lifestyle plays a crucial role in maintaining proper reproductive functions [6,7,19,20].

An unhealthy hypercaloric diet, excessive intake of saturated fats and trans-fatty acids, high glycaemic index, and low nutritional density may be directly associated with increased oxidative stress, which constitutes the underlying cause of obesity, intestinal dysbiosis, type 2 diabetes, and insulin resistance [21]. The above-mentioned metabolic disorders are associated with a deterioration of fertility mainly due to the generation of oxidative stress, regarded as one of the main factors leading to decreased sperm quality and a higher risk of infertility, as well as hormonal and immunological disorders [22]. Thus, an increase in white adipose tissue leads to an increase in the production of pro-inflammatory cytokines and reactive oxygen species, as well as the aromatase activity that is responsible for the conversion of testosterone to oestradiol. On the other hand, obese men with type 2 diabetes and insulin resistance are more likely to experience secondary hypogonadism and lower levels of sex hormone-binding protein (SHBG). Moreover, hyperglycaemia has a negative impact on sperm motility and the fertilization process [21,23,24,25,26].

Thus, nutritional intervention seems to be an extremely important element in the treatment of male infertility related to abnormal sperm parameters.

## 2. Nutritional Model Increasing the Risk of Male Infertility

In recent decades, the main nutritional model of developing and developed countries has become the so-called western diet [27,28]. As is presented in Figure 1, the western diet is characterized by a high intake of animal proteins, saturated and trans fatty acids, and simple carbohydrates, as well as a low supply of dietary fibre and essential unsaturated fatty acids (EFA). Additionally, it is a hypercaloric diet that is pro-inflammatory, with low nutritional density [28].

It is clear that with the spread of the western diet model, the parameters evaluating semen quality have deteriorated [29,30]. A diet rich in processed and according to some sources, red meat, fatty dairy, coffee, alcohol, sweet drinks and sweets, potatoes, and simultaneously deficient in whole-grain products, vegetables and fruits, poultry, fish and seafood, nuts, and lean dairy is associated with poorer semen parameters and reduced fertility [6,7,30,31,32]. Characteristics of a diet negatively affecting fertility and its proposed modifications are presented in Figure 1. 

A high-fat diet and obesity, promoted by an unhealthy lifestyle, affect the structure of spermatozoa, as well as the development of offspring and their health in later life. In fact, improper dietary patterns, such as meal omissions, insufficient intake of antioxidants, and high-energy density have been observed in infertile men [7].

### 2.1. Trans and Saturated Fatty Acids

It was demonstrated that spermatogenesis is negatively affected by trans-fatty acids and according to some sources, also saturated fats, which are rich in the western diet [6]. In fact, both polyunsaturated fatty acids (PUFA), as well as trans-fatty acids accumulate in the testes; however, unlike PUFA, the content of trans-fatty acids in semen and their consumption is associated with poorer sperm quality, as well as with lower sperm concentration in ejaculate [33,34]. What is more, animal studies suggest that a diet rich in trans fats may be associated with reduced testosterone production and testicular mass, as well as the initiation of pathological changes in the testes [32,35,36,37]. In fact, omega-6 fatty acids are also worth mentioning; in particular, if their supply is too high in relation to omega-3 fatty acids. They may also adversely affect fertility since they are likely to induce inflammation of a slight intensity, oxidative stress, dysfunction of the endothelium, and atherosclerosis [38]. 

A cross-sectional study conducted on a group of 209 healthy men indicates that the intake of trans and omega-6 fatty acids, as well as the reduction of omega-3 intake are associated with deterioration of testicular endocrine function, i.e., lower concentration levels of free testosterone and total testosterone, and with lower testicular volume. [36]. On the other hand, according to a cross-sectional study performed on 701 healthy men, the consumption of saturated fats results in lower sperm concentration in semen and lower semen count [39]. In both studies, semen and blood samples were collected from the participants. They were also asked to fill in questionnaires concerning their lifestyle, health, and diet. Both the participants in the Minguez-Alarcon et al. study, as well as 95% of the participants in the Jensen et al. study, did not know their fertility status, which ensured the objectivity of the study according to the authors [36,39].

The main sources of harmful fatty acids in the diet are fast-food products, salty and sweet snacks, ready-made confectionery, and processed and red meat [40].

### 2.2. Meat

According to the available research studies, the consumption of meat, especially processed meat, has a detrimental effect on fertility, which may stem from factors such as a high content of saturated fat and trans-fatty acids and the presence of preservatives and hormone residues [41,42]. It has been shown that processed red meat contains more residues of active substances that may affect the endocrine system than unprocessed meat [43]. The trans fatty acids present in meat may also affect sperm quality [44]. In the study by Afeiche et al., the consumption of red processed meat inversely correlated with the total number of spermatozoa in the ejaculate, as well as with the percentage of progressive sperm motility [45].

### 2.3. Smoking and Alcohol

Furthermore, it is vital to mention the use of stimulants. Researchers present a consistent view regarding the adverse effects of smoking on male fertility, both in terms of cigarettes and cannabis [46,47]. Occasional drinking of alcohol does not seem to have a negative effect on the quality of semen; however, daily alcohol consumption results in the deterioration of both semen volume and sperm morphology [48].

### 2.4. Caffeine

It has also been suggested that caffeine intake may impair male reproductive function, probably by means of triggering abnormalities in spermatozoa DNA. Nevertheless, most research studies do not demonstrate a link between moderate coffee consumption and male fertility. In the meta-analysis, covering 57 cross-sectional studies, which included 29,914 participants, no significant effect of coffee on sperm quality was found [46]. Interestingly, a review of 28 observational studies, which involved 19,967 men, suggested that caffeine from coffee, tea, and cocoa beverages did not have a negative impact on the quality of semen. In contrast, in a number of studies, sweet drinks containing caffeine were associated with decreased semen volume and count, as well as with lower sperm concentration [49].

### 2.5. Phytoestrogens

There are also certain controversies related to men’s reproductive health associated with phytoestrogens [6,50]. Phytoestrogens are compounds of plant origin, presenting a number of oestrogen-like activities. The best known groups of phytoestrogens are isoflavones, which in the form of ganistein and daidzein, are most abundant in soybeans and their preparations. It is suggested that isoflavones could potentially constitute an alternative to hormone replacement therapy in menopausal women. In addition to estrogenic effects, they present antimutagenic and antioxidant properties [51,52]. Furthermore, Asians possess a greater ability to convert soya to a non-steroidal oestrogen, i.e., as equol, by intestinal bacteria, which is influenced by both genetic conditionings and the diet, as well as the composition of intestinal microbiota [53]. The available studies do not demonstrate that moderate soybean intake has been associated with an increased risk of infertility, deterioration of semen quality, and a decrease in blood testosterone levels [50,54,55,56]. Moreover, in some cases, an improvement in sperm quality parameters was observed [57]. More studies describing the effect of isoflavones are necessary to clearly determine the effect of soya consumption on fertility. 

### 2.6. Contaminates

There is still no consistent approach with regard to compounds, such as bisphenol A or phthalates, i.e., ingredients in plastic food packaging. Currently, due to the small number of studies and their large limitations, there is not enough evidence to state that exposure to these substances at low or moderate levels has a negative effect on male fertility. Nevertheless, it seems reasonable to consider their adverse effects on reproductive health as possible [6,58,59]. 

However, the impact of pesticides and pollutants seems worth considering. Danielewicz et al. did not manage to prove that a pro-healthy diet model based on frequent consumption of fruit, vegetables, legumes, soups, properly composed meals, whole-grain products, juices, and nuts was associated with better sperm quality. The authors presumed that vegetables and fruits, which were rich in the diet, were also a source of pesticides and pollution. In fact, pesticides and insecticides were shown to have a greater impact on the deterioration of semen quality than the beneficial effects of microelements, vitamins, and antioxidants contained in vegetables and fruits [29]. This in turn suggests that it is vital to pay particular attention to a daily diet being based on products from reliable sources.

### 2.7. Hypercaloric Diet

In recent decades, a drastic change in society’s lifestyle has been observed concerning a reduction in energy expenditure, particularly in daily physical activity, increased consumption of hypercaloric foods with a high glycaemic index and high fat content, and a simultaneous low intake of dietary fibre [60]. This, in turn, has resulted in a significant increase in the proportion of obese individuals worldwide, which has since emerged as a global obesity pandemic [24,30,60,61]. More than half of Europeans are overweight or obese and men are much more likely to be overweight than women [62].

It is generally accepted that excessive body weight has a negative impact on the body, contributing to the development of diseases, such as diabetes, hypertension, cardiovascular diseases, cancer, sleep apnoea, or osteoarthritis. In fact, the impact of obesity on reproductive functions is also relevant [30,60,61].

Weight loss in obese men seems to be the first and most basic step in the treatment of male infertility.

## 3. Mechanisms Associating Improper Diet and Obesity with Infertility

Oxidative stress constitutes the key mechanism that associates improper diet and obesity with both lower semen quality and an increased risk of infertility. Moreover, it is currently considered one of the leading causes of male infertility [19], together with the decrease in antioxidant activity and dysfunctional activity of mitochondria in spermatozoa. Figure 2 presents the influence of oxidative stress on sperm quality and fertility.

Oxidative stress is reported to represent 30%–80% of male infertility cases [25,26]. Additionally, reactive oxygen species (ROS) may impair the motility of spermatozoa and interfere with their ability to connect to the oocyte [25].

As indicated in Figure 2, cell membrane lipids, proteins, and sperm DNA are damaged [20] once ROS overcomes the sperm antioxidant barrier. As a consequence, the higher the intensity of oxidative stress, the lower the motility, live sperm count, and sperm concentration in the semen, as well as the risk of miscarriage and child development abnormalities [2,22,25,63]. Moreover, excessive production of ROS is also associated with the deterioration of sperm morphology parameters [64]. 

Both the consumption of pro-inflammatory products and a low consumption of high antioxidant potential foods, as well as a high glycaemic index and load in the diet, constitute factors responsible for the increase in oxidative stress. Moreover, glucose metabolism proves to have significant impact on spermatogenesis, whereas hyperglycaemia affects sperm motility and the fertilization process [6].

In addition, in obese individuals, disorders on the hypothalamic-pituitary-gonads (HPG) axis have been observed. Excess fat tissue results in increased aromatase activity converting testosterone to oestrogen and consequently, to increased oestrogen levels with decreased levels of testosterone and LH hormone, which stimulates steroidogenesis, as well as of FSH, which is responsible for spermatogenesis [23,24,61,63].

Additionally, white adipose tissue (WAT) produces pro-inflammatory cytokines and ROS, the excess of which leads to systemic inflammation and oxidative stress [24]. One of the adipokines produced by WAT is leptin, also referred to as the satiety hormone. Under physiological conditions, leptin inhibits the appetite centre and stimulates the secretion of gonadotropic hormones [24,30]. The plasma concentration of leptin is proportional to the amount of body fat; however, in obese individuals, the phenomenon of leptin-resistance is observed next to hyperleptinemia, in which the hypothalamus does not respond properly to leptin. It is therefore suggested that excessive leptin production may constitute a significant factor in the development of androgen deficiency and deterioration of male reproductive function. Nevertheless, the mechanisms binding leptin to the HPG axis and obesity are still not fully understood and the research results on the role of leptin in infertility development remains unclear [65,66,67].

Apart from leptin, the hypothalamic-pituitary-gonads (HPG) axis and spermatogenesis are also affected by other pro-inflammatory cytokines produced in excess by the WAT, such as tumour necrosis factor (TNFa, tumor necrosis factor-a), interleukin-6, chemerin, resistin, or ghrelin [24].

Furthermore, excessive body weight is also associated with erectile dysfunction and an increased temperature in the scrotum, which may have an adverse effect on spermatogenesis. Additionally, it also affects the obstructive sleep apnoea syndrome, potentially leading to dysfunction of the HPG axis and disturbed night-time testosterone secretion due to chronic hypoxia and sleep fragmentation [24,63].

## 4. Intestinal Microbiota Disorders and Male Fertility

Recent studies have investigated the effects of intestinal microbiota on health. Intestinal microbiota comprises a group of microorganisms which inhabit the digestive tract. It consists of about 100 trillion microorganisms, functioning in a symbiotic and mutualistic relationship with the human organism. The dominant types of bacteria colonizing the adult gastrointestinal tract are Firmicutes, Bacteroidetes, Actinobacteria, Proteobacteria, Verrucomicrobia, and Fusobacteria [68]. The composition of intestinal microbiota is essential for many aspects of human health, including the immune system and predisposition to metabolic diseases [69]. 

The composition of intestinal microbiota largely depends on diet and may significantly change as a result of dietary modifications. An improper diet, characterized by a high intake of fat and monosaccharides, may lead to intestinal dysbiosis, i.e., quantitative and qualitative disturbance of intestinal microbiota composition, resulting in increased permeability of the intestinal barrier [70]. This, in turn, induces chronic inflammation in the body and is potentially at the root of disorders such as visceral disease, type 1 diabetes mellitus, inflammatory bowel disease, colorectal cancer, and obesity [71,72].

The researchers primarily emphasize that a high-fat diet may increase the amount of Mollicutes and Clostridum, belonging to the Firmicutes type, as well as of Bilophila and Enterobacteriaceae of Proteobacteria, and at the same time may contribute to the reduction of Bifidobacteria, Lactobacillus, Akkermansia muciniphila, and Bacteroidetes [73,74].

Bifidobacteria have the ability to modulate the intestinal barrier, reduce the concentration of lipopolysaccharide, and alleviate endotoxemia; in turn, Lactobacilus bacteria exhibit anti-inflammatory properties and facilitate the transport of short chain fatty acids (SCFA) [72].

It has been suggested that the adverse changes in the microbiota composition may also result from the lack of soluble fibre fractions in the diet. In contrast, polyphenols may contribute to the restoration of intestinal barrier integrity [75,76].

Ding et al. investigated the relationship between high-fat diet-induced dysbiosis (45% fat) in mice and spermatogenesis and sperm motility. In mice on a high-fat diet, a decrease in Bacteroidetes and Verrucomicrobia was observed, with an increase in Firmicutes and Proteobacteria. After faecal transplantation from mice fed on a high-fat diet to mice on a normal diet, a significant reduction in the number of sperm in the semen and a deterioration in sperm motility were noted, which indicates a possible effect of intestinal dysbiosis on fertility. Moreover, the number of Bacteroides and Prevotella increased significantly. It has been suggested that the spermatogenesis deterioration may be caused by elevated blood endotoxin levels, epididitis, and disturbances in gene expression in the testis [77].

This is the first research study regarding the influence of intestinal dysbiosis on sperm quality and spermatogenesis; therefore, it is essential to improve the existing knowledge on the relationship between intestinal microbiota and fertility.

## 5. A Dietary Model Supporting Male Fertility

Male semen is a mixture of secretions of different glands. It includes acid phosphatase, citric acid, inositol, copper, calcium, zinc and magnesium, fructose, seminogelin, vitamins C and E, prostaglandins, carnitine, glycerophosphato-choline, and neutral alpha-glucosidase. Additionally, sperm also consists of protein, carotenoids, electrolytes-sodium, and potassium, or glucose, selenium, urea, lactic acid, and cholesterol. Depending on the diet, the pH of sperm ranges from 7.2 to 8.2. Approximately 70% of the semen volume comprises secretions of seminal vesicles [25,78,79].

Many of these key components, which are essential for proper spermatogenesis and maturation of spermatozoa and their functioning, are sourced from food. Thus, insufficient supply in the diet may be crucial with regard to spermatogenesis, sperm quality, and male fertility [6,25,80]. According to the available research data, comparing the semen composition of men, a reduced content of zinc, magnesium, calcium, copper, and selenium was observed in infertile subjects as compared to men with normal fertility [81,82].

Research studies indicate that healthy dietary models clearly correlate with better sperm quality and a lower possibility of abnormalities in such parameters as sperm quantity, concentration, and motility, as well as with reduced sperm DNA fragmentation [6,7,20,26].

The recommended dietary standard is a diet rich in raw vegetables and fruit, whole-grain, and fibre-rich products, instead of products based on purified flour, which is indicated in Table 1. Olive oil, oily sea fish from a reliable source, nuts, seeds and stones, and avocados are good sources of unsaturated fats, which can make up to 35% of the calorific value in the diet. Therefore, a good source of protein is lean poultry and low-fat dairy products, legumes, fish, and seafood [6,83]. 

What is more, the role of selected minerals, antioxidant vitamins, and omega-3 fatty acids should be emphasized, the action of which will be based primarily on the minimization of oxidative stress. Furthermore, it seems promising to include carnitine and coenzyme Q10 supplements in the therapeutic intervention [84].

### 5.1. Mediterranean Diet

The Mediterranean Diet (MD) is considered to be a dietary model within the principles of a pro-fertility diet. MD is characterised by the consumption of large quantities of fruit and vegetables, wholemeal products, olive oil, nuts, and fish. Many health benefits of the Mediterranean diet have been demonstrated, mainly due to its antioxidant, anti-inflammatory, and lipid-reducing effects. In fact, this diet is recommended as a preventive measure against cardiovascular diseases, type 2 diabetes, neurodegenerative diseases, atrial fibrillation, and breast cancer [83,87]. The consumption of a MD has also been associated with better quality of semen in observational studies, but further interventional studies in this area are required to determine whether it may contribute to a higher chance of positive pregnancy outcomes [83,88,89,90].

### 5.2. Antioxidants for Male Infertility–What is the Evidence?

Although oxidative stress tends to be the primary factor underlying male infertility, it should be stressed that studies on the efficiency of antioxidant therapy are still contradictory. It seems that oral antioxidant supplementation improves the parameters evaluating semen quality and is associated with less DNA damage. However, no reliable studies evaluating the impact of antioxidant supplementation on positive pregnancy outcomes and live birth rate are available [91]. On the basis of 7 randomized studies involving the live birth rate following antioxidant supplementation, it was demonstrated that such supplementation in infertile men can improve the live birth rate. However, the authors of the review emphasize that the studies are ambiguous and subject to a high risk of error [92]. Thus, the need for more good quality research on efficacy, safety, and the recommended doses of antioxidants was also stressed in other research studies [92,93,94].

#### 5.2.1. Zinc

Zinc constitutes the basic element in the context of male fertility. Both seminal plasma and the prostate gland are characterized by its high content [95,96]. The appropriate level of zinc in semen is essential for the production of spermatozoa, preservation of their correct morphology, sperm count and function, and thus, for the proper course of fertilization. Moreover, testicular development and the proper course of steroidogenesis depend on zinc—the deficiency of this element is observed in patients with hypogonadism and underdeveloped secondary sexual traits, as well as in patients with oligospermia, astenozoospermia, and azoospermia [81,95,97,98].

Appropriate zinc concentrations in semen are associated with higher concentrations of spermatozoa in ejaculate, higher motility, viability, and increased antioxidant activity due to excessive amounts of superoxide anions by inhibiting nicotinamide adenine dinucleotide phosphate oxidase (NADPH oxidase) [99]. Zinc in the testis is crucial for spermatogenesis and the physiology of spermatozoa by maintaining the integrity of the genome in spermatozoa and correct structure [95,96]. Moreover, according to researchers, zinc is effective in protecting sperm from bacterial and chromosome damage. Due to strong antioxidant properties, an adequate amount of zinc in the semen plasma shows protective effects [95,100]. 

#### 5.2.2. Selenium

Another significant microelement is selenium, which is a component of glutathione peroxidase and thus increases the enzymatic antioxidant activity [20]. In several studies, lower selenium levels in the semen of infertile men were found in comparison to the healthy population. However, both the deficiency and the excess of selenium may result in fertility disorders and abnormal semen parameters [81,101,102,103,104,105]. Moreover, selenium has a protective effect against oxidative stress on sperm DNA and simultaneously increases motility and sperm viability [99]. In the course of normal spermatogenesis, apart from glutathione peroxidase, selenoprotein P is the key element [106]. In fact, the greatest amount of selenium occurs in the testis in this form.

#### 5.2.3. Other Antioxidants

In addition to selenium, vitamin C and tocopherol also present antioxidant properties by means of free radical neutralization. Therefore, it is important for the diet to be rich in vegetables and fruits, which are the main sources of these elements [107]. Furthermore, apart from its antioxidant properties, tocopherol is likely to have a protective effect against heavy metal damage [7,108]. Vegetables and fruits, especially raw green-leaved vegetables, are a source of folic acid, which is important in the course of spermatogenesis, particularly in the supplementation combined with zinc [109]. According to researchers, coenzyme Q10 may also be relevant in terms of semen quality, since in its reduced form, as ubiquinol and ubisemichinone radical, it has an antioxidant effect and is involved in all energy-dependent processes, including sperm motility. Remarkably, ubiquinone is capable of regenerating other antioxidants, such as vitamin C and vitamin E [110]. On the basis of the meta-analysis, patients receiving coenzyme Q10 showed a higher level of this substance in semen, as well as an increased concentration and better sperm motility compared to placebo [107]. However, it is possible that coenzyme Q10 delivered with the diet is not sufficient and does not result in the improvement of semen quality parameters—thus, supplementation is recommended [111,112].

What is more, the supplementation of L-carnitine may also be of importance [113,114,115]. It has been shown to have a positive impact on sperm maturation and motility and spermatogenesis in terms of providing energy supply to sperm by transporting long-chain fatty acids to mitochondria [20,84].

Lycopene, a powerful antioxidant belonging to the carotenoid family, seems to show promising results. It is indicated that lycopene reduces lipid peroxidation and DNA damage, strengthens the immune system, and increases the number and survival of sperm [116]. Zaremba et al. demonstrated a positive correlation between the consumption of lycopene and normal sperm morphology [117]. 

Many beneficial properties are also attributed to N-acetyl-cysteine (NAC), which is involved in glutathione synthesis (GSH) and has the ability to capture ROS [118]. The presence of NAC in the diet of infertile men has been associated with an increased number and motility of spermatozoa, as well as an increased number of normal structure spermatozoa following 3 months of supplementation. Moreover, a decrease in sperm DNA fragmentation and an increase in protamine levels, as well as a decrease in FSH and LH and an increase in testosterone levels in blood were observed [119]. 

A supplementation combining many antioxidants seems particularly beneficial. All the papers focused on this issue so far have demonstrated beneficial results of multiple antioxidants on sperm parameters [19]. For instance, Gvozdjáková et al. have shown that even a 3 month supplementation with carnitine, ubiquinol, and vitamins E and C has a positive effect on sperm density and motility. Additionally, the percentage of abnormal spermatozoa has also decreased [120]. It is worth emphasizing that the consumption of products with high antioxidant potential may also minimize the adverse effects of trans fats on sperm quality [45].

The most frequently used antioxidants, both in monotherapy and combined supplementation, include vitamin E and C, carnitine, coenzyme Q10, zinc, selenium, folic acid, and N-acetylcysteine [107].

#### 5.2.4. Omega-3 Fatty Acids

Omega-3 fatty acids, which are precursors to eicosanoids, are also known to have anti-inflammatory and antioxidant properties. Compared to other body tissues and cells, testis and spermatozoa have a higher concentration of polyunsaturated fatty acids and effective fertilization depends on the lipid composition of the sperm membrane [7,12]. It has been demonstrated that they positively affect the concentration and number and morphology of sperm and have the ability to modify the composition of the cell membrane by building into it, thus supporting its functioning [6,9,32,121]. It is also indicated that eicosapentaenoic acid (EPA) and/or docosahexaenoic acid (DHA) supplementation with fatty acids significantly increases sperm motility and DHA concentration in semen [122,123].

A meta-analysis of 16 randomized controlled trials showed a positive relationship between omega-3 supplementation and semen quality parameters in infertile men. Moreover, pro-healthy dietary models containing fish and seafood were also associated with better sperm quality in observational studies [123].

In addition, the consumption of 75 g of walnuts per day over a 12 week period was associated with a longer lifespan, motility, and sperm morphology. Interestingly, according to another study, the addition of 60 g of nut mixture to the Western diet, apart from improving the abovementioned parameters, also resulted in an increase in sperm count [85,86].

### 5.3. Magnesium, Calcium, Copper, Manganese

It is also recommended to provide an adequate supply of magnesium and calcium. The former constitutes a key element in the course of spermatogenesis and sperm motility, and also in the female reproductive tract. Furthermore, calcium affects the motility, hyperactivation, and capitulation of sperm and ultimately, the acrosome reaction, leading to sperm penetration into the oocyte [81]. In addition, copper is also necessary for the proper functioning of sperm and manganese affects the motility of sperm and the fertilization process [124,125,126,127]. Nevertheless, both manganese and copper in excessive amounts show an adverse effect on sperm [128,129]. The knowledge about the impact of selected trace elements and vitamins on male semen is summarized in Figure 3.

### 5.4. Fibre

On the basis of the research studies, the role of fibre should also be emphasized. It is an essential element of the diet with regard to fertility due to its potential binding of non-conjugated oestrogens, which is directly associated with lower levels of oestrogen in plasma. In fact, its appropriate levels are necessary to maintain proper reproductive functions [6].

## 6. Summary

Nutrition can both negatively and positively affect the quality of semen. The diet should include vegetables and fruit, fish and seafood, nuts, seeds, whole-grain and fibre-rich products, poultry, and low-fat dairy products. On the other hand, low consumption of fruit and vegetables and products with an antioxidant potential, a high calorific intake, a diet rich in saturated fatty acids and trans fats, low fish consumption, as well as a high proportion of both red and processed meat have a negative impact on the quality of semen, which may contribute to reduced male fertility.

Therefore, a modification of lifestyle, particularly with regard to the diet, seems to be indispensable with regard to male infertility associated with semen quality.

## Figures and Tables

**Figure 1 jcm-09-01400-f001:**
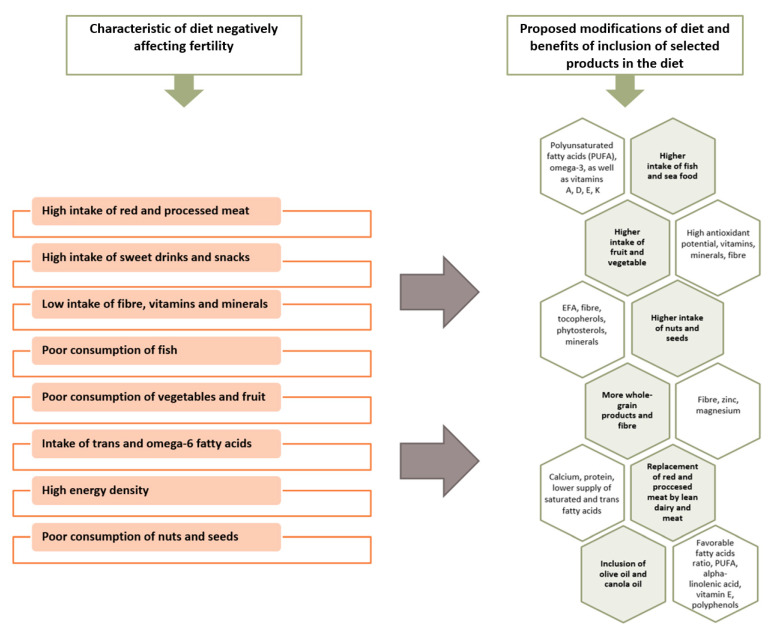
Characteristics of a diet negatively affecting fertility and its proposed modifications. [6,7,31].

**Figure 2 jcm-09-01400-f002:**
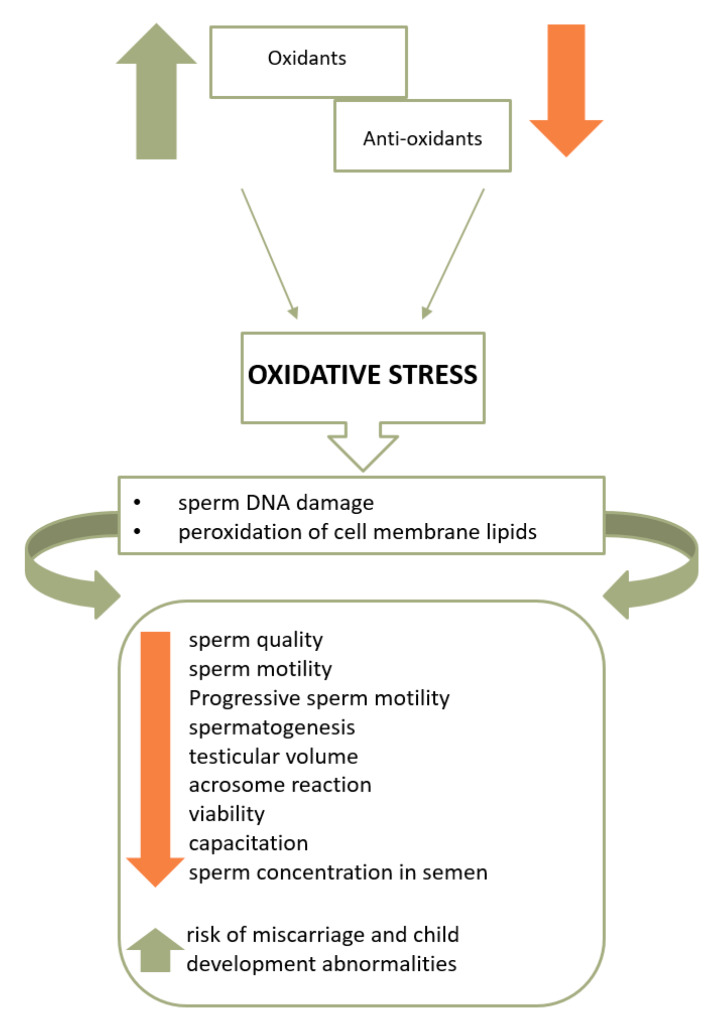
The influence of oxidative stress on sperm quality and fertility [11,65].

**Figure 3 jcm-09-01400-f003:**
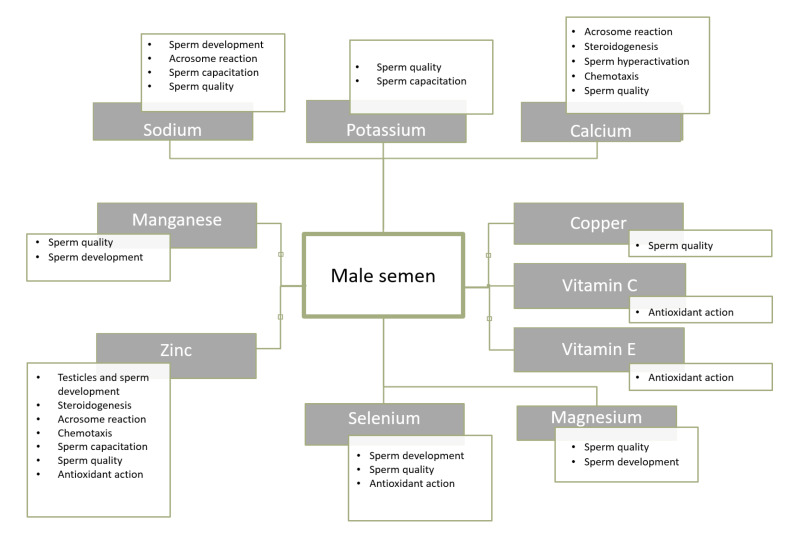
Selected components of male semen and their role [81].

**Table 1 jcm-09-01400-t001:** Characteristics of a diet beneficial for fertility.

Dietary Component/Items	Active Substances	Comments/Remarks
Oily sea fish	PUFA, omega-3 Fat-soluble vitamins A, D, E, K	Fish and seafood represent the main sources of DHA and EPA in the diet, therefore their incorporation in the diet may be associated with the improvement of semen quality [44]. Fish are often contaminated with mercury and other neurotoxic substances [44].
Vegetables and fruit	Antioxidants, folic acid, fibre, minerals	Vegetables and fruits provide the basis for pro-healthy nutrition models, which are associated with the improvement of semen quality and fertility [6,7,20,25,83]. It is worth choosing raw vegetables and fruits. Research suggests that pesticide residues may modify the beneficial effect of fruit and vegetable consumption on the quality of semen [29].
Nuts, seeds	EFAs, fibre, tocopherols, phytosterols, polyphenols, minerals	It is important to choose nuts and unroasted and unsalted seeds. The use of nuts in the diet may have a beneficial effect on the quality of sperm [84,85,86].
Whole-grain products	Fibre, zinc, magnesium	It is recommended to limit the consumption of refined flour products and choose whole-grain products, which are rich in fibre [6,20].
Lean dairy	Calcium, a wholesome protein	It is beneficial to choose low-fat dairy products, due to a lower saturated fat content [20].
Olive oil, rapeseed oil	PUFA, alpha-linolenic acid, vitamin E, polyphenols	It is advisable to substitute saturated fats with vegetable oils containing unsaturated acid residues [6,7].

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
