# Peer review of "Diet and Nutritional Factors in Male (In)fertility—Underestimated Factors"

_jcm, 2020, doi:10.3390/jcm9051400_

Round 1

Reviewer 1 Report

Authors performed a narrative review to highlight how diet and nutritional factors can influence sperm quality, and though can condition male fertility status. In particular, it emphasizes the positive role of antioxidants, since oxidative stress is currently considered one of the leading cause of male infertility, together with the decrease in antioxidant activity and a dysfunctional activity of mitochondria in spermatozoa.

The review is exhaustive and well written, but several issues should be addressed:

Major COMMENTS:

  • Lines 56-59 introduce one of the key point of this revision. This paragraph should be developed in more details which appropriate references.
  • Figure 1 is not clear to me. It is clear the left part of the figure, but not the modification that this figure should represents. I suggest to better explain the figure in the figure legend.
  • Lines 81-87: the possible mechanisms by which spermatogenesis is damaged by PUFA should be reported and described in more details.
  • The main body of the text should be divided in paragraph, reporting evidences for each nutrient considered. For example, it should be created a paragraph for meat, coffee, pesticides and so on.
  • Lines 172-174. The authors reported two contrasting sentences on the role of leptin in testosterone secretion. Please, revise or explain the reason for this discrepancy.

MINOR COMMENT

  • Authors remark the importance of anti-oxidants in meliorating semen quality and semen parameters. So why didn’t they use “anti-oxidants” as a key word?
  • Line 36: The authors stated: “Among the causes of male infertility, the most common are oligospermia…” it is not correct to define oligospermia the causes of male infertility. Oligospermia, such as other alteration detected during semen analysis should be considered clinical characteristics of infertile men. Not a cause, but a feature.
  • Lines 48-52: there are many other studies, more recent, evaluating the effect of environment on sperm parameters.
  • Lines 87-93: a discussion about the limits and strengths of these clinical studies could be useful to understand the strength of these findings. Moreover,: why did authors use “in contrast” to compare the results of the two different cross-sectional studies? They get to different conclusions that are not “in contrast”, they just highlight the two sides of the same coin. Authors should use “on the other hand”.
  • Lines 183-189: A brief introduction, explaining the physiological role and composition of microbiota could be useful.
  • A dietary model supporting male fertility, line 294: authors directly used acronyms “EPA” and “DHA” without specifying that they were effectively referring to eicosapentaenoic acid and to docosahexaenoic acid. Please revise.
  • Authors always used the British English spelling for “fibre”, except in the Figure 1, left side, where they used the American spelling “fiber”.

Author Response

REVIEWER 1.

Open Review

English language and style

( ) Extensive editing of English language and style required
( ) Moderate English changes required
(x) English language and style are fine/minor spell check required
( ) I don't feel qualified to judge about the English language and style

Comments and Suggestions for Authors

Authors performed a narrative review to highlight how diet and nutritional factors can influence sperm quality, and though can condition male fertility status. In particular, it emphasizes the positive role of antioxidants, since oxidative stress is currently considered one of the leading cause of male infertility, together with the decrease in antioxidant activity and a dysfunctional activity of mitochondria in spermatozoa.

The review is exhaustive and well written, but several issues should be addressed:

REPLY:

Thank you for any comments and suggestions of this review, we will try to point to them point by point, answer all questions and make corrections that will increase the value of the submitted work.

The manuscript has been language-corrected by a professional biomedical translation company TranslationLab.

Major COMMENTS:

Comment: Lines 56-59 introduce one of the key point of this revision. This paragraph should be developed in more details which appropriate references.

Reply: Thank you for this suggestion. We developed this paragraph with more details and appropriate references as below.

An unhealthy, hypercaloric diet, excessive intake of saturated fats and trans-fatty acids, high glycaemic index and low nutritional density may be directly associated with increased oxidative stress, which constitutes the underlying cause of obesity, intestinal dysbiosis, type 2 diabetes and insulin resistance [21]. The above-mentioned metabolic disorders are associated with a deterioration of fertility mainly due to the generation of oxidative stress, regarded as one of the main factors leading to a decreased sperm quality and a higher risk of infertility, as well as to hormonal and immunological disorders [29]. Thus, an increase in white adipose tissue leads to an increase in the production of pro-inflammatory cytokines and reactive oxygen species, as well as in the aromatase activity responsible for the conversion of testosterone to oestradiol. On the other hand, obese men with type 2 diabetes and insulin resistance are more likely to experience secondary hypogonadism and lower levels of sex hormone-binding protein (SHBG). Moreover, hyperglycaemia has a negative impact on sperm motility and the fertilization process [21-25].

We also added new references:

  1. Varani, J. Healthful Eating, the Western Style Diet and Chronic Disease. APDV 2017, 1.
  2. Chambers, T.; Anderson, R. The impact of obesity on male fertility. HJ 2015.

Comment: Figure 1 is not clear to me. It is clear the left part of the figure, but not the modification that this figure should represents. I suggest to better explain the figure in the figure legend.

Reply: Thank you for this valuable suggestion. We have made some changes in Figure 1 to make it more clear and understable.

Figure 1. Characteristics of a diet negatively affecting fertility and its proposed modifications [6,7,30].

Comment: Lines 81-87: the possible mechanisms by which spermatogenesis is damaged by PUFA should be reported and described in more details.

Reply: Thank you for that comment and suggestion. We would like to precise that in our statement we did not suggest the negative impact of PUFA on spermatogenesis. We pointed out that PUFA, especially omega-3, may affect spermatogenesis but in positive way. The wording of the sentence may have been unclear, which is why we have introduced changes to the paragraph. We also added some informations about omega-6 fatty acids, which in excess can have negative impact on health.

It was demonstrated that spermatogenesis is negatively affected by trans-fatty acids and, according to some sources, also saturated fats, which are rich in the western diet [6]. In fact, both PUFA, as well as trans-fatty acids, accumulate in the testes; however, unlike PUFA, the content of trans-fatty acids in the semen and their consumption is associated with poorer sperm quality, as well as with lower sperm concentration in the ejaculate [33,34]. What is more, animal studies suggest that a diet rich in trans fats may be associated with reduced testosterone production and testicular mass, as well as the initiation of pathological changes in the testes [31,35-37]. In fact, omega-6 fatty acids are also worth mentioning; in particular, if their supply is too high in relation to omega-3 fatty acids. They may also adversely affect fertility since they are likely to induce inflammation of slight intensity, oxidative stress, dysfunction of the endothelium and atherosclerosis [38].

Added references:

  1. DiNicolantonio, J.J.; O’Keefe, J.H. Importance of maintaining a low omega–6/omega–3 ratio for reducing inflammation. Open Heart 2018, 5, e000946.

Comment: The main body of the text should be divided in paragraph, reporting evidences for each nutrient considered. For example, it should be created a paragraph for meat, coffee, pesticides and so on.

Reply: Thank you for that valuable suggestion. According to your comment we divided main body of paragraph and the work seems to be more readable and comprehensible.

  1. Nutritional model increasing the risk of male infertility

In recent decades, the main nutritional model of the developing and developed countries has become the so-called western diet [26,27]. As it is presented in Figure 1, western diet is characterized by a high intake of animal proteins, saturated and trans fatty acids and simple carbohydrates, as well as a low supply of dietary fibre and essential unsaturated fatty acids (EFA). Additionally, it is a hypercaloric diet with low nutritional density and pro-inflammatory character [27].

It is clear that with the spread of the Western diet model, the parameters evaluating semen quality have deteriorated [28,32]. A diet rich in processed and, according to some sources, red meat, fatty dairy, coffee, alcohol, sweet drinks and sweets, potatoes, and simultaneously deficient in whole-grain products, vegetables and fruits, poultry, fish and seafood, nuts, and lean dairy is associated with poorer semen parameters and reduced fertility [6,7,30,32]. Characteristics of a diet negatively affecting fertility and its proposed modifications are presented in Figure 1.

Figure 1. Characteristics of a diet negatively affecting fertility and its proposed modifications. [6,7,30].

A high-fat diet and obesity, promoted by an unhealthy lifestyle, affect the structure of spermatozoa, as well as the development of offspring and their health in later life. In fact, improper dietary patterns, such as meal omissions, insufficient intake of antioxidants and high-energy density have been observed in infertile men [7].

2.1. Trans and saturated fatty acids

It was demonstrated that spermatogenesis is negatively affected by trans-fatty acids and, according to some sources, also saturated fats, which are rich in the western diet [6]. In fact, both PUFA, as well as trans-fatty acids, accumulate in the testes; however, unlike PUFA, the content of trans-fatty acids in the semen and their consumption is associated with poorer sperm quality, as well as with lower sperm concentration in the ejaculate [33,34]. What is more, animal studies suggest that a diet rich in trans fats may be associated with reduced testosterone production and testicular mass, as well as the initiation of pathological changes in the testes [31,35-37]. In fact, omega-6 fatty acids are also worth mentioning; in particular, if their supply is too high in relation to omega-3 fatty acids. They may also adversely affect fertility since they are likely to induce inflammation of slight intensity, oxidative stress, dysfunction of the endothelium and atherosclerosis [38].

Cross-sectional study conducted on a group of 209 healthy men indicates that the intake of trans and omega-6 fatty acids, as well as the reduction of omega-3 intake are associated with a deterioration of testicular endocrine function, i.e. lower concentration levels of free testosterone and total testosterone, and with lower testicular volume. [36]. On the other hand, according to the cross-sectional study performed on 701 healthy men, the consumption of saturated fats results in lower sperm concentration in semen and lower semen count [39]. In both studies, semen and blood samples were collected from the participants. They were also asked to fill in questionnaires concerning their lifestyle, health and diet. Both the participants in the Minguez-Alarcon et al. study, as well as 95% of the participants in the Jensen et al. study, did not know their fertility status, which ensured the objectivity of the study according to the authors [36,39].

     The main sources of harmful fatty acids in the diet are fast-food products, salty and sweet snacks, ready-made confectionery and processed and red meat [40].

2.3. Meat

According to the available research studies, the consumption of meat, especially processed meat, has a detrimental effect on fertility, which may stem from, such factors as high content of saturated fat and trans-fatty acids, the presence of preservatives and hormone residues [41,42]. It has been shown that red processed meat contains more residues of active substances that may affect the endocrine system than unprocessed meat [43]. The trans fatty acids present in meat may also affect sperm quality [44]. In the study by Afeiche et al., the consumption of red processed meat inversely correlated with the total number of spermatozoa in the ejaculate, as well as with the percentage of progressive sperm motility [45].

2.4. Smoking and alcohol

Furthermore, it is vital to mention the use of stimulants. Researchers present a consistent view regarding the adverse effects of smoking on male fertility, both in terms of cigarettes and cannabis [46,47]. Occasional drinking of alcohol does not seem to have a negative effect on the quality of semen; however, daily alcohol consumption results in the deterioration of both semen volume and sperm morphology [48].

2.5. Caffeine

It has also been suggested that caffeine intake may impair male reproductive function, probably by means of triggering abnormalities in spermatozoa DNA. Nevertheless, most research studies do not demonstrate a link between moderate coffee consumption and male fertility. In the meta-analysis, covering 57 cross-sectional studies which included 29914 participants, no significant effect of coffee on sperm quality was found [46]. Interestingly, a review of 28 observational studies, which involved 19 967 men, suggested that caffeine from coffee, tea and cocoa beverages did not have a negative impact on the quality of semen. In contrast, in a number of studies, sweet drinks containing caffeine were associated with a decreased semen volume and count, as well as with lower sperm concentration [49].

2.4 Phytoestrogens

There are also certain controversies related to men's reproductive health associated with phytoestrogens [6,50]. Phytoestrogens are compounds of plant origin, presenting a number of oestrogen-like activities. The best known groups of phytoestrogens are isoflavones, which in the form of ganistein and daidzein are most abundant in soybean and its preparations. It is suggested that isoflavones could potentially constitute an alternative to hormone replacement therapy in menopausal women. In addition to estrogenic effects, they present antimutagenic and antioxidant properties [51,52]. Furthermore, Asians possess a greater ability to convert soya to a non-steroidal oestrogen, i.e. as equol, by intestinal bacteria, which is influenced by both the genetic conditionings and the diet, as well as the composition of intestinal microbiota [53]. The available studies do not demonstrate that moderate soybean intake has been associated with an increased risk of infertility, deterioration of semen quality, and a decrease in blood testosterone levels [50, 54-56]. Moreover, in some cases an improvement in sperm quality parameters was observed [57]. More studies describing the effect of isoflavones are necessary to clearly determine the effect of soya consumption on fertility.

2.6. Contaminates

There is still no consistent approach with regard to compounds, such as bisphenol A or phthalates, i.e. ingredients in plastic food packaging. Currently, due to the small number of studies and their large limitations, there is not enough evidence to state that exposure to these substances at low or moderate levels has a negative effect on male fertility. Nevertheless, it seems reasonable to consider their adverse effects on reproductive health as possible [6, 59-60].

However, the impact of pesticides and pollutants seems worth considering. Danielewicz et al. did not manage to prove that a pro-healthy diet model based on frequent consumption of fruit, vegetables, legumes, soups, properly composed meals, whole-grain products, juices and nuts was associated with better sperm quality. The authors presumed that vegetables and fruits, which were rich in diet, were also a source of pesticides and pollution. In fact, pesticides and insecticides were shown to have a greater impact on the deterioration of semen quality than the beneficial effects of microelements, vitamins and antioxidants contained in vegetables and fruits [28]. This in turn suggests that it is vital to pay particular attention for a daily diet to be based on products from reliable sources.

2.7. Hypercaloric diet

In recent decades a drastic change in society's lifestyle has been observed concerning a reduction in energy expenditure, particularly in daily physical activity, the consumption of hypercaloric foods with high glycaemic index and high fat content has increased, with a simultaneous low intake of dietary fibre [60]. This, in turn, has resulted in a significant increase in the proportion of obese individuals worldwide - which has since emerged as a global obesity pandemic [23,32,60-61]. More than half of Europeans are overweight or obese, and men are much more likely to be overweight than women [62].

It is generally accepted that excessive body weight has a negative impact on the body, contributing to the development of diseases, such as diabetes, hypertension, cardiovascular diseases, cancer, sleep apnoea or osteoarthritis. In fact, the impact of obesity on reproductive functions is also relevant [32,60,61].

Weight loss in obese men seems to be the first, and the most basic step in the treatment of male infertility.

  1. A dietary model supporting male fertility

Male semen is a mixture of secretions of different glands. It includes acid phosphatase, citric acid, inositol, copper, calcium, zinc and magnesium, fructose, seminogelin, vitamins C and E, prostaglandins, carnitine, glycerophosphato-choline and neutral alpha-glucosidase. Additionally, sperm also consists of protein, carotenoids, electrolytes - sodium and potassium, or glucose, selenium, urea, lactic acid and cholesterol. Depending on the diet, the pH of the sperm ranges from 7.2 to 8.2. Approximately 70% of the semen volume comprises secretions of seminal vesicles. [24,78,79]

Many of these key components, essential for proper spermatogenesis, maturation of spermatozoa and their functioning, have their source in food. Thus, their insufficient supply in the diet may be crucial with regard to spermatogenesis, sperm quality and male fertility [6,24,80]. According to the available research data, comparing the semen composition of men, a reduced content of zinc, magnesium, calcium, copper and selenium was observed in infertile subjects as compared to men with normal fertility [81,82]. 

Research studies indicate that healthy dietary models clearly correlate with better sperm quality and lower possibility of abnormalities in such parameters as sperm quantity, concentration and motility, as well as with reduced sperm DNA fragmentation [6,7,20,25].

The recommended dietary standard is a diet rich in raw vegetables and fruit, whole-grain and fibre-rich products, instead of products based on purified flour, which is indicated in Table 1. Olive oil, oily sea fish from a reliable source, nuts, seeds and stones, avocados are good sources of unsaturated fats, which can make up to 35 % of the calorific value in the diet. Therefore, a good source of protein will be lean poultry and low-fat dairy products, legumes, fish and seafood [6, 108].

What is more, the role of selected minerals, antioxidant vitamins and omega-3 fatty acids should be emphasized, the action of which will be based primarily on the minimization of oxidative stress. Furthermore, it seems promising to include carnitine and coenzyme Q10 supplements in the therapeutic intervention [83].

Table 1. Characteristics of a diet beneficial for fertility

Dietary component / items

Active substances

Comments / Remarks

Oily sea fish

PUFA, omega-3

Fat-soluble vitamins - A, D, E, K

Fish and seafood represent the main sources of DHA and EPA in the diet, therefore their incorporation in the diet may be associated with the improvement of semen quality [44]. Fish are often contaminated with mercury and other neurotoxic substances [44].

Vegetables and fruit

Antioxidants, folic acid, fibre, minerals

Vegetables and fruits provide the basis for pro-healthy nutrition models, associated with the improvement of semen quality and fertility [6,7,20,25,84]. It is worth choosing raw vegetables and fruits.

Research suggests that pesticide residues may modify the beneficial effect of fruit and vegetable consumption on the quality of semen [28].

Nuts, seeds

EFAs, fibre, tocopherols, phytosterols, polyphenols, minerals

It is important to choose nuts and unroasted and unsalted seeds. The use of nuts in the diet may have a beneficial effect on the quality of sperm[110,111].

Whole-grain products

Fibre, zinc, magnesium

It is recommended to limit the consumption of refined flour products, and choose whole-grain products, rich in fibre [6,20].

Lean dairy

Calcium, a wholesome protein

It is beneficial to choose low-fat dairy products, due to a lower saturated fat content [20].

Olive oil, rapeseed oil

PUFA, alpha-linolenic acid, vitamin E, polyphenols

It is advisable to substitute saturated fats with vegetable oils containing unsaturated acid residues [6,7].

5.1. Mediterranean diet

The Mediterranean Diet (MD) is considered to be a dietary model within the principles of a pro-fertility diet. MD is characterised by the consumption of large quantities of fruit and vegetables, wholemeal products, olive oil, nuts and fish. Many health benefits of the Mediterranean diet have been demonstrated, mainly due to its antioxidant, anti-inflammatory and lipid-reducing effects. In fact, this diet is recommended as a preventive measure against cardiovascular diseases, type 2 diabetes, neurodegenerative diseases, atrial fibrillation and breast cancer [108,114]. The consumption of MDs has also been associated with a better quality of semen in observational studies, but further interventional studies in this area are required to determine whether it may contribute to a higher chance of positive pregnancy outcomes [108,112,113, 115].    

5.2. Antioxidants for male infertility – what is the evidence?

Although oxidative stress tends to be the primary factor underlying male infertility, it should be stressed that studies on the efficiency of antioxidant therapy are still contradictory. It seems that oral antioxidant supplementation improves the parameters evaluating semen quality and is associated with less DNA damage. However, no reliable studies evaluating the impact of antioxidant supplementation on positive pregnancy outcomes and live birth rate are available [116]. On the basis of 7 randomized studies, involving the live birth rate following antioxidant supplementation, it was demonstrated that such supplementation in infertile men can improve the live birth rate. However, the authors of the review emphasize that the studies are ambiguous and subject to high risk of error [117]. Tus, the need for more good quality research on efficacy, safety and the recommended doses of antioxidants was also stressed in other research studies [117-119].

5.2.1. Zinc

Zinc constitutes the basic element in the context of male fertility. Both seminal plasma and the prostate gland are characterized by its high content [84,85]. Appropriate level of zinc in semen is essential for the production of spermatozoa, preservation of their correct morphology, sperm count and function, and thus for the proper course of fertilization. Moreover, testicular development and the proper course of steroidogenesis depend on zinc – the deficiency of this element is observed in patients with hypogonadism and underdeveloped secondary sexual traits, as well as in patients with oligospermia, astenozoospermia and azoospermia [81,84,86,87].

Appropriate zinc concentrations in the semen are associated with higher concentration of spermatozoa in the ejaculate, higher motility, viability and increased antioxidant activity due to excessive amount of superoxide anions by inhibiting NADPH oxidase [93]. Zinc in the testis is crucial for spermatogenesis and the physiology of spermatozoa by maintaining the integrity of the genome in spermatozoa and correct structure [84,85]. Moreover, according to researchers, zinc is effective in protecting sperm from bacterial and chromosome damage. Due to strong antioxidant properties, an adequate amount of zinc in the semen plasma shows protective effects [84,119].

5.2.2. Selenium

Another significant microelement is selenium, which is a component of glutathione peroxidase and thus increases the enzymatic antioxidant activity [20]. In several studies lower selenium levels in semen of infertile men were found in comparison to the healthy population. However, both the deficiency and the excess of selenium may result in fertility disorders and abnormal semen parameters [81,89-92]. Moreover, selenium has a protective effect against oxidative stress on sperm DNA, and simultaneously increases motility and sperm viability [93]. In the course of normal spermatogenesis, apart from glutathione peroxidase, selenoprotein P is the key element. In fact, the greatest amount of selenium occurs in the testis in this form.

5.2.3. Other antioxidants

In addition to selenium, vitamin C and tocopherol also present antioxidant properties by means of free radical neutralization. Therefore, it is important for the diet to be rich in vegetables and fruits, which are the main sources of these elements [88]. Furthermore, apart from its antioxidant properties, tocopherol is likely to have a protective effect against heavy metal damage [7,96]. Vegetables and fruits, especially raw green-leaved vegetables are a source of folic acid, which is important in the course of spermatogenesis, particularly in the supplementation combined with zinc [97]. According to researchers, also coenzyme Q10 may be relevant in terms of semen quality, since in its reduced form, as ubiquinol and ubisemichinone radical, it has an antioxidant effect, and is involved in all energy-dependent processes, including sperm motility. Remarkably, ubiquinone is capable of regenerating other antioxidants, such as vitamin C and vitamin E [98]. On the basis of the meta-analysis, patients receiving coenzyme Q10 showed a higher level of this substance in semen, as well as an increased concentration and better sperm motility compared to placebo [88]. However, it is possible that coenzyme Q10 delivered with the diet is not sufficient and does not result in improvement of semen quality parameters – thus, supplementation is recommended [99,100].

What is more, the supplementation of L-carnitine may also be of importance [101-103]. It has been shown to have a positive impact on sperm maturation and motility and spermatogenesis in terms of providing energy supply to sperm by transporting long-chain fatty acids to mitochondria [20,83].

Lycopene, a powerful antioxidant belonging to the carotenoid family, seems to show promising results. It is indicated that lycopene reduces lipid peroxidation and DNA damage, strengthens the immune system and increases the number and survival of sperm [120]. Zaremba et al. demonstrated a positive correlation between the consumption of lycopene and normal sperm morphology [121].

 Many beneficial properties are also attributed to N-acetyl-cysteine (NAC), which is involved in glutathione synthesis (GSH) and has the ability to capture ROS [122]. The presence of NAC in the diet of infertile men has been associated with an increased number and motility of spermatozoa, as well as an increased number of normal structure spermatozoa following 3 months of supplementation. Moreover, a decrease in sperm DNA fragmentation and an increase in protamine levels, as well as a decrease in FSH, LH and an increase in testosterone levels in blood were observed [123].

A supplementation combining many antioxidants seems particularly beneficial. All the papers focused on this issue so far have demonstrated beneficial results of multiple antioxidants on sperm parameters [19]. For instance, Gvozdjáková et al. have shown that even a 3-month supplementation with carnitine, ubiquinol and vitamins E and C has a positive effect on sperm density and motility. Additionally, the percentage of abnormal spermatozoa has also decreased [98]. It is worth emphasizing that the consumption of products with high antioxidant potential may also minimize the adverse effects of trans fats on sperm quality [45].

The most frequently used antioxidants, both in monotherapy and combined supplementation, include vitamin E and C, carnitine, coenzyme Q10, zinc, selenium, folic acid and N-acetylcysteine [88].

5.2.4. Omega-3 fatty acids

Omega-3 fatty acids, which are precursors to eicosanoids, are also known to have anti-inflammatory and antioxidant properties. Compared to other body tissues and cells, testis and spermatozoa have a higher concentration of polyunsaturated fatty acids, and effective fertilization depends on the lipid composition of the sperm membrane [7,104]. It has been demonstrated that they positively affect the concentration, number and morphology of sperm and have the ability to modify the composition of the cell membrane by building into it, thus, supporting its functioning [6,9,31,105]. It is also indicated that eicosapentaenoic acid (EPA) and/or docosahexaenoic acid (DHA) supplementation with fatty acids significantly increases sperm motility and DHA concentration in semen [106].

A meta-analysis of 16 randomized controlled trials showed a positive relationship between omega-3 supplementation and semen quality parameters in infertile men. Moreover, pro-healthy dietary models containing fish and seafood were also associated with better sperm quality in observational studies [107].

In addition, the consumption of 75g of walnuts per day over a 12 week period was associated with a longer lifespan, motility and sperm morphology. Interestingly, according to another study, the addition of 60g of nut mixture to the Western diet, apart from improving the abovementioned parameters, also resulted in an increase in sperm count [110,111].

5.3. Magnesium, calcium, copper, manganese

It is also recommended to provide an adequate supply of magnesium and calcium. The former  constitutes a key element in the course of spermatogenesis and sperm motility, also in the female reproductive tract. Furthermore, calcium affects the motility, hyperactivation and capitulation of sperm and ultimately the acrosome reaction leading to sperm penetration into the oocyte [81]. In addition, copper is also necessary for the proper functioning of sperm, and manganese affects the motility of sperm and the fertilization process [124-127]. Nevertheless, both manganese and copper in excessive amounts show an adverse effect on sperm [128,129]. The knowledge about the impact of selected trace elements and vitamins on male semen is summarized in Figure 3.

Figure 3. Selected components of male semen and their role [81].

5.4. Fibre

On the basis of the research studies, the role of fibre should also be emphasized. It is an essential element of the diet with regard to fertility due to its potential binding of non-conjugated oestrogens, which is directly associated with lower levels of oestrogen in plasma. In fact, its appropriate levels are necessary to maintain proper reproductive functions [6].

Comment: Lines 172-174. The authors reported two contrasting sentences on the role of leptin in testosterone secretion. Please, revise or explain the reason for this discrepancy.

    Reply: Thank you for that important remark. We explained inaccuracy in manuscript.

One of the adipokines produced by WAT is leptin, also referred to as the satiety hormone. Under physiological conditions, leptin inhibits the appetite centre and stimulates the secretion of gonadotropic hormones [32,23]. The plasma concentration of leptin is proportional to the amount of body fat; however, in obese individuals, the phenomenon of leptin-resistance is observed next to hyperleptinemia, in which the hypothalamus does not respond properly to leptin. It is therefore suggested that excessive leptin production may constitute a significant factor in the development of androgen deficiency and deterioration of male reproductive function. Nevertheless, the mechanisms binding leptin to the HPG axis and obesity are still not fully understood, and the research results on the role of leptin in infertility development remains unclear [66,67].

Added references: 

  1. Khodamoradi, K.; Parmar, M.; Khosravizadeh, Z.; Kuchakulla, M.; Manoharan, M.; Arora, H. The role of leptin and obesity on male infertility: Current Opinion in Urology 2020, 30, 334–339.

  1. Zhang, J.; Gong, M. Review of the role of leptin in the regulation of male reproductive function. Andrologia 2018, 50, e12965.

MINOR COMMENT

Comment: Authors remark the importance of anti-oxidants in meliorating semen quality and semen parameters. So why didn’t they use “anti-oxidants” as a key word?

Reply: Thank you for that comment.

According to your suggestion we added „anti-oxidants” as a key word.

Comment: Line 36: The authors stated: “Among the causes of male infertility, the most common are oligospermia…” it is not correct to define oligospermia the causes of male infertility. Oligospermia, such as other alteration detected during semen analysis should be considered clinical characteristics of infertile men. Not a cause, but a feature.

Reply: Thank you for that comment. We totally agree that the irregularities we have mentioned are not causes but the characteristics of male infertility. According to your comment we have made correction.

There are several characteristic features of male infertility, such as oligospermia, i.e. low sperm concentration in semen, asthenozoospermia, i.e. an absolute lack of motility, or a decreased motility of spermatozoa, and teratozoospermia, i.e. an insufficient number of spermatozoa of normal structure [7].

Comment: Lines 48-52: there are many other studies, more recent, evaluating the effect of environment on sperm parameters.

Reply: Thank you for that comment.

 We added as below:

Environmental factors which significantly affect male fertility comprise smoking cigarettes and cannabis, anabolic steroid use, excessive alcohol consumption, emotional stress, excessive exposure to high temperatures, age, tight clothing, environmental pollution, sedentary lifestyle, exposure to pesticides and toxins, radiofrequency electromagnetic radiation, as well as cytotoxic drugs, cadmium and lead [1,6,10-18]. It is vital to bear in mind that some factors such as age, environmental pollution or radiation cannot be avoided [12,16]. However, some research studies suggest that the use of antioxidants, such as resveratrol, may constitute a therapeutic alternative [17].

Acoording to your suggestion we added more recent, valuable studies.

  1. Gabrielsen, J.S.; Tanrikut, C.; Chronic exposures and male fertility: the impacts of environment, diet, and drug use on spermatogenesis. Andrology 2016, 4, 648-61.
  2. Walczak-Jędrzejowska, R. Oxidative Stress and Male Infertility. Part I: Factors causing oxidative stress in semen. Advances in Andrology Online 2015, 2, 5-15.
  3. Chiang, C.; Mahalingam, S.; Flaws, J. Environmental Contaminants Affecting Fertility and Somatic Health. Semin Reprod Med 2017, 35, 241–249.
  4. Christou, M.A.; Christou, P.A.; Markozannes, G.; Tsatsoulis, A.; Mastorakos, G.; Tigas, S. Effects of Anabolic Androgenic Steroids on the Reproductive System of Athletes and Recreational Users: A Systematic Review and Meta-Analysis. Sports Med 2017, 47, 1869–1883.
  5. Kesari, K.K.; Agarwal, A.; Henkel, R. Radiations and male fertility. Reprod Biol Endocrinol 2018, 16, 118.
  6. Sansone, A.; Di Dato, C.; de Angelis, C.; Menafra, D.; Pozza, C.; Pivonello, R.; Isidori, A.; Gianfrilli, D. Smoke, alcohol and drug addiction and male fertility. Reprod Biol Endocrinol 2018, 16, 3.
  7. Hart, K.; Tadros, N.N. The role of environmental factors and lifestyle on male reproductive health, the epigenome, and resulting offspring. Panminerva Med 2019, 61.
  8. Alamo, A.; Condorelli, R.A.; Mongioì, L.M.; Cannarella, R.; Giacone, F.; Calabrese, V.; La Vignera, S.; Calogero, A.E. Environment and Male Fertility: Effects of Benzo-α-Pyrene and Resveratrol on Human Sperm Function In Vitro. JCM 2019, 8, 561.
  9. Duca, Y.; Aversa, A.; Condorelli, R.A.; Calogero, A.E.; La Vignera, S. Substance Abuse and Male Hypogonadism. JCM 2019, 8, 732.

Comment: Lines 87-93: a discussion about the limits and strengths of these clinical studies could be useful to understand the strength of these findings. Moreover,: why did authors use “in contrast” to compare the results of the two different cross-sectional studies? They get to different conclusions that are not “in contrast”, they just highlight the two sides of the same coin. Authors should use “on the other hand”.

Reply: Thank you for that comment. As you suggested, we added brief discussion about the limits and strenghts of studies.

In both studies, semen and blood samples were collected from the participants. They were also asked to fill in questionnaires concerning their lifestyle, health and diet. Both the participants in the Minguez-Alarcon et al. study, as well as 95% of the participants in the Jensen et al. study, did not know their fertility status, which ensured the objectivity of the study according to the authors [36,39].

According to your suggestion we also changed „in contrast” to „on the other hand”, as you can see below:

On the other hand, according to the cross-sectional study performed on 701 healthy men, the consumption of saturated fats results in lower sperm concentration in semen and lower semen count [39].

Comment: Lines 183-189: A brief introduction, explaining the physiological role and composition of microbiota could be useful.

Reply: Thank you for this suggestion. We have included introduction to the paragraph as you suggested.

Recent studies have investigated the effects of intestinal microbiota on health. Intestinal microbiota comprises a group of microorganisms which inhabit the digestive tract. It consists of about 100 trillion microorganisms, functioning in symbiotic and mutualistic relationship with human organism. The dominant types of bacteria colonizing the adult gastrointestinal tract are Firmicutes, Bacteroidetes, Actinobacteria, Proteobacteria, Verrucomicrobia, and Fusobacteria [68]. The composition of intestinal microbiota is essential for many aspects of the human health, including the immune system and predisposition to metabolic diseases [69].

Added references:

  1. Valdes, A. M.; Walter, J.; Segal, E.; Spector, T.D. Role of the gut microbiota in nutrition and health. BMJ 2018, 361, k2179.
  2. Nagpal, R.; Kumar, M.; Yadav, A.K.; Hemalatha, R.; Yadav, H.; Marotta, F.; Yamashiro, Y. Gut microbiota in health and disease: an overview focused on metabolic inflammation. Beneficial Microbes 2017, 7, 181–194.

Comment: A dietary model supporting male fertility, line 294: authors directly used acronyms “EPA” and “DHA” without specifying that they were effectively referring to eicosapentaenoic acid and to docosahexaenoic acid. Please revise.

Reply: Thank you for that remark. We've added an explanation for the acronyms.

It is also indicated that eicosapentaenoic acid (EPA) and/or docosahexaenoic acid (DHA) supplementation with fatty acids significantly increases sperm motility and DHA concentration in semen [106].

Comment: Authors always used the British English spelling for “fibre”, except in the Figure 1, left side, where they used the American spelling “fiber”.

Reply: Thank you for that comment.

We have made correction in Figure 1 as you can see below. (pdf)

Reviewer 2 Report

The ms contains a review of literature data on effect of diet on male fertility. Only minor changes are suggested.

1.Consider changing “ western lifestyle” to „lifestyle”, „testicle” to „testis”

  1. “zinc is effective in protecting sperm from bacterial and chromosome damage” Rewrite or give more details
  2. Effect of phytoestrogen should be included; Mediterranean diet , Asian paradox

Author Response

REVIEWER 2.

Open Review

English language and style

( ) Extensive editing of English language and style required
( ) Moderate English changes required
( ) English language and style are fine/minor spell check required
(x) I don't feel qualified to judge about the English language and style

Comments and Suggestions for Authors

The ms contains a review of literature data on effect of diet on male fertility. Only minor changes are suggested.

REPLY:

Thank you for any comments and suggestions of this review, we will try to point to them point by point, answer all questions and make corrections that will increase the value of the submitted work. 

The manuscript has been language-corrected by a professional biomedical translation company TranslationLab.

Comment: Consider changing “ western lifestyle” to „lifestyle”, „testicle” to „testis”

Reply: Thank you for this comment. According to your suggestion we have made corrections in our work. We changed “western lifestyle” to “unhealthy lifestyle” because we want to emphasize that only improper lifestyle patterns negatively affect fertility. We also changed “testicles” to “testis” as you recommended. Thank you for your valuable suggestions.

Our changes:

A high-fat diet and obesity, the development of which is encouraged by the unhealthy lifestyle, affects the structure of spermatozoa, but also the development of the offspring and their health in later stages of life.

A high-fat diet and obesity, resulting from the unhealthy lifestyle, affects the structure of spermatozoa, but also the development of the offspring and their health in later stages of life.

In fact, both PUFA, as well as trans-fatty acids, accumulate in the testes; however, unlike PUFA, the content of trans-fatty acids in the semen and their consumption is associated with poorer sperm quality, as well as with lower sperm concentration in the ejaculate [33,34]. What is more, animal studies suggest that a diet rich in trans fats may be associated with reduced testosterone production and testicular mass, as well as the initiation of pathological changes in the testes [31,35-37].

Comment: “zinc is effective in protecting sperm from bacterial and chromosome damage” Rewrite or give more details.

Reply: Thank you for this comment. According to your suggestion we added brief explanation. We also added one confirming reference.

Moreover, according to researchers, zinc is effective in protecting sperm from bacterial and chromosome damage. Due to strong antioxidant properties, an adequate amount of zinc in the semen plasma shows protective effects [84, 120].

We added reference:

  1. Gammoh, N.Z.; Rink, L. Zinc in Infection and Inflammation. Nutrients 2017, 25.

Comment: Effect of phytoestrogen should be included; Mediterranean diet, Asian paradox.

    REPLY: Thank you for this suggestion. We haveve added a section on the health effects of phytoestrogen, including the Asian paradox.

There are also certain controversies related to men's reproductive health associated with phytoestrogens [6,50]. Phytoestrogens are compounds of plant origin, presenting a number of oestrogen-like activities. The best known groups of phytoestrogens are isoflavones, which in the form of ganistein and daidzein are most abundant in soybean and its preparations. It is suggested that isoflavones could potentially constitute an alternative to hormone replacement therapy in menopausal women. In addition to estrogenic effects, they present antimutagenic and antioxidant properties [51,52]. Furthermore, Asians possess a greater ability to convert soya to a non-steroidal oestrogen, i.e. as equol, by intestinal bacteria, which is influenced by both the genetic conditionings and the diet, as well as the composition of intestinal microbiota [53]. The available studies do not demonstrate that moderate soybean intake has been associated with an increased risk of infertility, deterioration of semen quality, and a decrease in blood testosterone levels [50, 54-56]. Moreover, in some cases an improvement in sperm quality parameters was observed [57]. More studies describing the effect of isoflavones are necessary to clearly determine the effect of soya consumption on fertility.

Added references:

  1. Cooper, A.R. To eat soy or to not eat soy: the ongoing look at phytoestrogens and fertility. Fertility and Sterility 2019, 112, 825–826.
  2. Messina, M.; Messina, V. The Role of Soy in Vegetarian Diets. Nutrients 2010, 2, 855–888.
  3. Křížová, L.; Dadáková, K.; Kašparovská, J.; Kašparovský, T. Isoflavones. Molecules 2019, 24, 1076.
  4. Desmawati, D.; Sulastri, D. A Phytoestrogens and Their Health Effect. OAMJMS 2019, 7, 495–499.
  5. Mitchell J.H.; Cawood E.; Kinniburgh D.; Provan A.; Collins A.R.; Irvine D.S. Effect of a phytoestrogen food supplement on reproductive health in normal males. Clin. Sci. 2001, 100, 613–618.
  6. Beaton L.K., McVeigh B.L., Dillingham B.L., Lampe J.W., Duncan A.M. Soy protein isolates of varying isoflavone content do not adversely affect semen quality in healthy young men. Fertil. Steril. 2009, 94, 1717-1722.
  7. Messina M., Watanabe S., Setchell K.D. Report on the 8th International Symposium on the Role of Soy in Health Promotion and Chronic Disease Prevention and Treatment. J. Nutr. 2009, 139, 796S–802S.
  8. Casini M.L., Gerli S., Unfer V. An infertile couple suffering from oligospermia by partial sperm maturation arrest: can phytoestrogens play a therapeutic role? A case report study. Gynecol. Endocrinol. 2006, 22, 399–401.

We also included information that the Mediterranean diet is a food model that meets the conditions for a fertile diet as below:

The Mediterranean Diet (MD) is considered to be a dietary model within the principles of a pro-fertility diet. MD is characterised by the consumption of large quantities of fruit and vegetables, wholemeal products, olive oil, nuts and fish. Many health benefits of the Mediterranean diet have been demonstrated, mainly due to its antioxidant, anti-inflammatory and lipid-reducing effects. In fact, this diet is recommended as a preventive measure against cardiovascular diseases, type 2 diabetes, neurodegenerative diseases, atrial fibrillation and breast cancer [108,114]. The consumption of MDs has also been associated with a better quality of semen in observational studies, but further interventional studies in this area are required to determine whether it may contribute to a higher chance of positive pregnancy outcomes [108,112,113,115].    

We added new references:

  1. Salas-Huetos, A.; Babio, N.; Carrell, D.T.; Bulló, M.; Salas-Salvadó, J. Adherence to the Mediterranean diet is positively associated with sperm motility: a cross-sectional analysis. Sci. Rep. 2019, 9, 3389.
  2. Ricci, E.; Bravi, F.; Noli, S.; Ferrari, S.; De Cosmi, V.; La Vecchia, I.; Cavadini, M.; La Vecchia, C.; Parazzini, F. Mediterranean diet and the risk of poor semen quality: cross‐sectional analysis of men referring to an Italian Fertility Clinic. Andrology 2019, 7, 156–162.
  3. Karayiannis, D.; Kontogianni, M.D.; Mendorou, C.; Douka, L.; Mastrominas, M.; Yiannakouris, N. Association between adherence to the Mediterranean diet and semen quality parameters in male partners of couples attempting fertility. Hum. Reprod. 2016, humrep;dew288v1.
  4. Tosti, V.; Bertozzi, B.; Fontana, L. Health Benefits of the Mediterranean Diet: Metabolic and Molecular Mechanisms. The Journals of Gerontology: Series A 2018, 73, 318–326
  5. Ricci, E.; Bravi, F.; Noli, S.; Somigliana, E.; Cipriani, S.; Castiglioni, M.; Chiaffarino, F.; Vignali, M.; Gallotti, B.; Parazzini, F. Mediterranean diet and outcomes of assisted reproduction: an Italian cohort study. American Journal of Obstetrics and Gynecology 2019, 221, 627.e1-627.e14.

Reviewer 3 Report

Abstract

The first sentence is difficult to understand. Should say something along the lines of "Infertility in couples is due solely to male factor in up to 50% of cases"

Introduction

Overall, the organization is very poor. Paragraphs are not constructed well and and do not flow well. Many paragraphs are one sentence. 

It should discuss more about diet and nutrition giving background of what is available. 

Body

 In line 36 the writing is not clear. The way oligospermia, asthenozoospermia and teratozoospermia are defined are not done in the same grammatical way. One uses ie., another uses a hyphen etc. 

In sentence 39 what does Leaver refer to?

In reference to diet and its impact on fertility the authors should discuss the actual nutritional content in the specified foods that is impactful. Generalizing categories of food does not help to understand the implications they have as many foods fall into them. 

The title " 2. Poor nutritional model of male fertility" does not accurately state the topic. The syntax is improper here, Poor refers to the quality of the model as used. Please use another title. 

The utility of Figure 1 is not clear. In the manuscript it says "Characteristics of a diet negatively  affecting fertility and its proposed modifications is presented in Figure 1." The figure does not demonstrate clearly at all how modifications are proposed. This figure should clearly show which foods are negatively impacting fertility.

In table 1 it would be useful to make a column on how each diet modification has been shown to actually improve fertility. 

The authors discuss antioxidant supplements and promote them in a bright light as possible therapeutic options. They should fully cover this topic and discuss the evidence and actual efficacy that has been shown behind these. There are several studies in the literature evaluating the evidence behind popular supplement products. 

Author Response

REVIEWER 3.

Open Review

English language and style

( ) Extensive editing of English language and style required
(x) Moderate English changes required
( ) English language and style are fine/minor spell check required
( ) I don't feel qualified to judge about the English language and style

Comments and Suggestions for Authors

REPLY:

Thank you for any comments and suggestions of this review, we will try to point to them point by point, answer all questions and make corrections that will increase the value of the submitted work. The manuscript has been language-corrected by a professional biomedical translation company TranslationLab.

Abstract

Comment: The first sentence is difficult to understand. Should say something along the lines of "Infertility in couples is due solely to male factor in up to 50% of cases"

Reply: Thank you for this valuable suggestion. We changed the first sentence as you suggested. It seems to be much more clear and understable.

In up to 50% of cases infertility issues stem solely from the male factor.

Introduction

Comment: Overall, the organization is very poor. Paragraphs are not constructed well and and do not flow well. Many paragraphs are one sentence. 

It should discuss more about diet and nutrition giving background of what is available. 

Reply: Thank you very much for this suggestion. According to your and other reviewers suggestions, we have added broader description of the relationship between diet and fertility and we have changed organization. We added as below:

  1. Introduction

Infertility, i.e. the inability to get pregnant, despite a regular, minimum yearly sexual intercourse without using any contraceptive, affects an increasing proportion of society [1-5].

It is estimated that as much as 15%, i.e. about 70 million couples in the world's reproductive age, experience problems with getting pregnant, with approximately half of the cases related to male infertility [2,4,6]. It is reported that an estimated 35% of infertility cases involve only women, 20% both women and men, 30% involve problems only on the part of the man, and 15% of infertility cases remain unexplained [1].

There are several characteristic features of male infertility, such as oligospermia, i.e. low sperm concentration in semen, asthenozoospermia, i.e. an absolute lack of motility, or a decreased motility of spermatozoa, and teratozoospermia, i.e. an insufficient number of spermatozoa of normal structure [7]. Leaver points out that these disorders constitute over 90% of male infertility causes [1]. According to an extensive meta-analysis covering 185 studies, including over 40 000 men from the developed countries, the number of spermatozoa, i.e. the main factor determining the quality of semen, decreased by 50-60% over the period 1973-2011 [8]. According to research carried out in Poland on a group of 169 young, healthy men with unknown fertility status from the Lower Silesia region, the average and median of 7 parameters determining sperm quality were within the limits of the WHO standards in 2010. However, sperm viability was close to the lower range of the norm, whereas the average percentage of abnormal sperm structure was as high as 85%. Nearly 9% of the studied cases had one, two or three parameters outside the limits of the standard [9].

Environmental factors which significantly affect male fertility comprise smoking cigarettes and cannabis, anabolic steroid use, excessive alcohol consumption, emotional stress, excessive exposure to high temperatures, age, tight clothing, environmental pollution, sedentary lifestyle, exposure to pesticides and toxins, radiofrequency electromagnetic radiation, as well as cytotoxic drugs, cadmium and lead [1,6,10-18]. It is vital to bear in mind that some factors such as age, environmental pollution or radiation cannot be avoided [12,16]. However, some research studies suggest that the use of antioxidants, such as resveratrol, may constitute a therapeutic alternative [17].

Furthermore, recent research data point to the fact that also diet is directly associated to semen quality, and that the overall lifestyle plays a crucial role in maintaining proper reproductive functions [6,7,19,20].

An unhealthy, hypercaloric diet, excessive intake of saturated fats and trans-fatty acids, high glycaemic index and low nutritional density may be directly associated with increased oxidative stress, which constitutes the underlying cause of obesity, intestinal dysbiosis, type 2 diabetes and insulin resistance [21]. The above-mentioned metabolic disorders are associated with a deterioration of fertility mainly due to the generation of oxidative stress, regarded as one of the main factors leading to a decreased sperm quality and a higher risk of infertility, as well as to hormonal and immunological disorders [29]. Thus, an increase in white adipose tissue leads to an increase in the production of pro-inflammatory cytokines and reactive oxygen species, as well as in the aromatase activity responsible for the conversion of testosterone to oestradiol. On the other hand, obese men with type 2 diabetes and insulin resistance are more likely to experience secondary hypogonadism and lower levels of sex hormone-binding protein (SHBG). Moreover, hyperglycaemia has a negative impact on sperm motility and the fertilization process [21-25].

Thus, nutritional intervention seems to be an extremely important element in the treatment of male infertility related to abnormal sperm parameters.

Body

Comment: In line 36 the writing is not clear. The way oligospermia, asthenozoospermia and teratozoospermia are defined are not done in the same grammatical way. One uses ie., another uses a hyphen etc. 

Reply: Thank you for that comment. We have changed definitions according to your suggestion.

There are several characteristic features of male infertility, such as oligospermia, i.e. low sperm concentration in semen, asthenozoospermia, i.e. an absolute lack of motility, or a decreased motility of spermatozoa, and teratozoospermia, i.e. an insufficient number of spermatozoa of normal structure [7].

Comment: In sentence 39 what does Leaver refer to?

Reply: Thank you for that comment. Leaver in monography refers to University of Maryland Medical Center.

Comment: In reference to diet and its impact on fertility the authors should discuss the actual nutritional content in the specified foods that is impactful. Generalizing categories of food does not help to understand the implications they have as many foods fall into them. 

Reply: Thank you for that comment. It’s very valuable suggestion.

According to your and other teviewers comments, we have divided as recommended so it’s seems to be more readable and understandable. You can see corrections below:

  1. Nutritional model increasing the risk of male infertility

In recent decades, the main nutritional model of the developing and developed countries has become the so-called western diet [26,27]. As it is presented in Figure 1, western diet is characterized by a high intake of animal proteins, saturated and trans fatty acids and simple carbohydrates, as well as a low supply of dietary fibre and essential unsaturated fatty acids (EFA). Additionally, it is a hypercaloric diet with low nutritional density and pro-inflammatory character [27].

It is clear that with the spread of the Western diet model, the parameters evaluating semen quality have deteriorated [28,32]. A diet rich in processed and, according to some sources, red meat, fatty dairy, coffee, alcohol, sweet drinks and sweets, potatoes, and simultaneously deficient in whole-grain products, vegetables and fruits, poultry, fish and seafood, nuts, and lean dairy is associated with poorer semen parameters and reduced fertility [6,7,30,32]. Characteristics of a diet negatively affecting fertility and its proposed modifications are presented in Figure 1.

Figure 1. Characteristics of a diet negatively affecting fertility and its proposed modifications. [6,7,30].

A high-fat diet and obesity, promoted by an unhealthy lifestyle, affect the structure of spermatozoa, as well as the development of offspring and their health in later life. In fact, improper dietary patterns, such as meal omissions, insufficient intake of antioxidants and high-energy density have been observed in infertile men [7].

2.1. Trans and saturated fatty acids

It was demonstrated that spermatogenesis is negatively affected by trans-fatty acids and, according to some sources, also saturated fats, which are rich in the western diet [6]. In fact, both PUFA, as well as trans-fatty acids, accumulate in the testes; however, unlike PUFA, the content of trans-fatty acids in the semen and their consumption is associated with poorer sperm quality, as well as with lower sperm concentration in the ejaculate [33,34]. What is more, animal studies suggest that a diet rich in trans fats may be associated with reduced testosterone production and testicular mass, as well as the initiation of pathological changes in the testes [31,35-37]. In fact, omega-6 fatty acids are also worth mentioning; in particular, if their supply is too high in relation to omega-3 fatty acids. They may also adversely affect fertility since they are likely to induce inflammation of slight intensity, oxidative stress, dysfunction of the endothelium and atherosclerosis [38].

Cross-sectional study conducted on a group of 209 healthy men indicates that the intake of trans and omega-6 fatty acids, as well as the reduction of omega-3 intake are associated with a deterioration of testicular endocrine function, i.e. lower concentration levels of free testosterone and total testosterone, and with lower testicular volume. [36]. On the other hand, according to the cross-sectional study performed on 701 healthy men, the consumption of saturated fats results in lower sperm concentration in semen and lower semen count [39]. In both studies, semen and blood samples were collected from the participants. They were also asked to fill in questionnaires concerning their lifestyle, health and diet. Both the participants in the Minguez-Alarcon et al. study, as well as 95% of the participants in the Jensen et al. study, did not know their fertility status, which ensured the objectivity of the study according to the authors [36,39].

     The main sources of harmful fatty acids in the diet are fast-food products, salty and sweet snacks, ready-made confectionery and processed and red meat [40].

2.3. Meat

According to the available research studies, the consumption of meat, especially processed meat, has a detrimental effect on fertility, which may stem from, such factors as high content of saturated fat and trans-fatty acids, the presence of preservatives and hormone residues [41,42]. It has been shown that red processed meat contains more residues of active substances that may affect the endocrine system than unprocessed meat [43]. The trans fatty acids present in meat may also affect sperm quality [44]. In the study by Afeiche et al., the consumption of red processed meat inversely correlated with the total number of spermatozoa in the ejaculate, as well as with the percentage of progressive sperm motility [45].

2.4. Smoking and alcohol

Furthermore, it is vital to mention the use of stimulants. Researchers present a consistent view regarding the adverse effects of smoking on male fertility, both in terms of cigarettes and cannabis [46,47]. Occasional drinking of alcohol does not seem to have a negative effect on the quality of semen; however, daily alcohol consumption results in the deterioration of both semen volume and sperm morphology [48].

2.5. Caffeine

It has also been suggested that caffeine intake may impair male reproductive function, probably by means of triggering abnormalities in spermatozoa DNA. Nevertheless, most research studies do not demonstrate a link between moderate coffee consumption and male fertility. In the meta-analysis, covering 57 cross-sectional studies which included 29914 participants, no significant effect of coffee on sperm quality was found [46]. Interestingly, a review of 28 observational studies, which involved 19 967 men, suggested that caffeine from coffee, tea and cocoa beverages did not have a negative impact on the quality of semen. In contrast, in a number of studies, sweet drinks containing caffeine were associated with a decreased semen volume and count, as well as with lower sperm concentration [49].

2.4 Phytoestrogens

There are also certain controversies related to men's reproductive health associated with phytoestrogens [6,50]. Phytoestrogens are compounds of plant origin, presenting a number of oestrogen-like activities. The best known groups of phytoestrogens are isoflavones, which in the form of ganistein and daidzein are most abundant in soybean and its preparations. It is suggested that isoflavones could potentially constitute an alternative to hormone replacement therapy in menopausal women. In addition to estrogenic effects, they present antimutagenic and antioxidant properties [51,52]. Furthermore, Asians possess a greater ability to convert soya to a non-steroidal oestrogen, i.e. as equol, by intestinal bacteria, which is influenced by both the genetic conditionings and the diet, as well as the composition of intestinal microbiota [53]. The available studies do not demonstrate that moderate soybean intake has been associated with an increased risk of infertility, deterioration of semen quality, and a decrease in blood testosterone levels [50, 54-56]. Moreover, in some cases an improvement in sperm quality parameters was observed [57]. More studies describing the effect of isoflavones are necessary to clearly determine the effect of soya consumption on fertility.

2.6. Contaminates

There is still no consistent approach with regard to compounds, such as bisphenol A or phthalates, i.e. ingredients in plastic food packaging. Currently, due to the small number of studies and their large limitations, there is not enough evidence to state that exposure to these substances at low or moderate levels has a negative effect on male fertility. Nevertheless, it seems reasonable to consider their adverse effects on reproductive health as possible [6, 59-60].

However, the impact of pesticides and pollutants seems worth considering. Danielewicz et al. did not manage to prove that a pro-healthy diet model based on frequent consumption of fruit, vegetables, legumes, soups, properly composed meals, whole-grain products, juices and nuts was associated with better sperm quality. The authors presumed that vegetables and fruits, which were rich in diet, were also a source of pesticides and pollution. In fact, pesticides and insecticides were shown to have a greater impact on the deterioration of semen quality than the beneficial effects of microelements, vitamins and antioxidants contained in vegetables and fruits [28]. This in turn suggests that it is vital to pay particular attention for a daily diet to be based on products from reliable sources.

2.7. Hypercaloric diet

In recent decades a drastic change in society's lifestyle has been observed concerning a reduction in energy expenditure, particularly in daily physical activity, the consumption of hypercaloric foods with high glycaemic index and high fat content has increased, with a simultaneous low intake of dietary fibre [60]. This, in turn, has resulted in a significant increase in the proportion of obese individuals worldwide - which has since emerged as a global obesity pandemic [23,32,60-61]. More than half of Europeans are overweight or obese, and men are much more likely to be overweight than women [62].

It is generally accepted that excessive body weight has a negative impact on the body, contributing to the development of diseases, such as diabetes, hypertension, cardiovascular diseases, cancer, sleep apnoea or osteoarthritis. In fact, the impact of obesity on reproductive functions is also relevant [32,60,61].

Weight loss in obese men seems to be the first, and the most basic step in the treatment of male infertility.

  1. A dietary model supporting male fertility

Male semen is a mixture of secretions of different glands. It includes acid phosphatase, citric acid, inositol, copper, calcium, zinc and magnesium, fructose, seminogelin, vitamins C and E, prostaglandins, carnitine, glycerophosphato-choline and neutral alpha-glucosidase. Additionally, sperm also consists of protein, carotenoids, electrolytes - sodium and potassium, or glucose, selenium, urea, lactic acid and cholesterol. Depending on the diet, the pH of the sperm ranges from 7.2 to 8.2. Approximately 70% of the semen volume comprises secretions of seminal vesicles. [24,78,79]

Many of these key components, essential for proper spermatogenesis, maturation of spermatozoa and their functioning, have their source in food. Thus, their insufficient supply in the diet may be crucial with regard to spermatogenesis, sperm quality and male fertility [6,24,80]. According to the available research data, comparing the semen composition of men, a reduced content of zinc, magnesium, calcium, copper and selenium was observed in infertile subjects as compared to men with normal fertility [81,82]. 

Research studies indicate that healthy dietary models clearly correlate with better sperm quality and lower possibility of abnormalities in such parameters as sperm quantity, concentration and motility, as well as with reduced sperm DNA fragmentation [6,7,20,25].

The recommended dietary standard is a diet rich in raw vegetables and fruit, whole-grain and fibre-rich products, instead of products based on purified flour, which is indicated in Table 1. Olive oil, oily sea fish from a reliable source, nuts, seeds and stones, avocados are good sources of unsaturated fats, which can make up to 35 % of the calorific value in the diet. Therefore, a good source of protein will be lean poultry and low-fat dairy products, legumes, fish and seafood [6, 108].

What is more, the role of selected minerals, antioxidant vitamins and omega-3 fatty acids should be emphasized, the action of which will be based primarily on the minimization of oxidative stress. Furthermore, it seems promising to include carnitine and coenzyme Q10 supplements in the therapeutic intervention [83].

Table 1. Characteristics of a diet beneficial for fertility

Dietary component / items

Active substances

Comments / Remarks

Oily sea fish

PUFA, omega-3

Fat-soluble vitamins - A, D, E, K

Fish and seafood represent the main sources of DHA and EPA in the diet, therefore their incorporation in the diet may be associated with the improvement of semen quality [44]. Fish are often contaminated with mercury and other neurotoxic substances [44].

Vegetables and fruit

Antioxidants, folic acid, fibre, minerals

Vegetables and fruits provide the basis for pro-healthy nutrition models, associated with the improvement of semen quality and fertility [6,7,20,25,84]. It is worth choosing raw vegetables and fruits.

Research suggests that pesticide residues may modify the beneficial effect of fruit and vegetable consumption on the quality of semen [28].

Nuts, seeds

EFAs, fibre, tocopherols, phytosterols, polyphenols, minerals

It is important to choose nuts and unroasted and unsalted seeds. The use of nuts in the diet may have a beneficial effect on the quality of sperm[110,111].

Whole-grain products

Fibre, zinc, magnesium

It is recommended to limit the consumption of refined flour products, and choose whole-grain products, rich in fibre [6,20].

Lean dairy

Calcium, a wholesome protein

It is beneficial to choose low-fat dairy products, due to a lower saturated fat content [20].

Olive oil, rapeseed oil

PUFA, alpha-linolenic acid, vitamin E, polyphenols

It is advisable to substitute saturated fats with vegetable oils containing unsaturated acid residues [6,7].

5.1. Mediterranean diet

The Mediterranean Diet (MD) is considered to be a dietary model within the principles of a pro-fertility diet. MD is characterised by the consumption of large quantities of fruit and vegetables, wholemeal products, olive oil, nuts and fish. Many health benefits of the Mediterranean diet have been demonstrated, mainly due to its antioxidant, anti-inflammatory and lipid-reducing effects. In fact, this diet is recommended as a preventive measure against cardiovascular diseases, type 2 diabetes, neurodegenerative diseases, atrial fibrillation and breast cancer [108,114]. The consumption of MDs has also been associated with a better quality of semen in observational studies, but further interventional studies in this area are required to determine whether it may contribute to a higher chance of positive pregnancy outcomes [108,112,113,115].    

5.2. Antioxidants for male infertility – what is the evidence?

Although oxidative stress tends to be the primary factor underlying male infertility, it should be stressed that studies on the efficiency of antioxidant therapy are still contradictory. It seems that oral antioxidant supplementation improves the parameters evaluating semen quality and is associated with less DNA damage. However, no reliable studies evaluating the impact of antioxidant supplementation on positive pregnancy outcomes and live birth rate are available [116]. On the basis of 7 randomized studies, involving the live birth rate following antioxidant supplementation, it was demonstrated that such supplementation in infertile men can improve the live birth rate. However, the authors of the review emphasize that the studies are ambiguous and subject to high risk of error [117]. Tus, the need for more good quality research on efficacy, safety and the recommended doses of antioxidants was also stressed in other research studies [117-119].

5.2.1. Zinc

Zinc constitutes the basic element in the context of male fertility. Both seminal plasma and the prostate gland are characterized by its high content [84,85]. Appropriate level of zinc in semen is essential for the production of spermatozoa, preservation of their correct morphology, sperm count and function, and thus for the proper course of fertilization. Moreover, testicular development and the proper course of steroidogenesis depend on zinc – the deficiency of this element is observed in patients with hypogonadism and underdeveloped secondary sexual traits, as well as in patients with oligospermia, astenozoospermia and azoospermia [81,84,86,87].

Appropriate zinc concentrations in the semen are associated with higher concentration of spermatozoa in the ejaculate, higher motility, viability and increased antioxidant activity due to excessive amount of superoxide anions by inhibiting NADPH oxidase [93]. Zinc in the testis is crucial for spermatogenesis and the physiology of spermatozoa by maintaining the integrity of the genome in spermatozoa and correct structure [84,85]. Moreover, according to researchers, zinc is effective in protecting sperm from bacterial and chromosome damage. Due to strong antioxidant properties, an adequate amount of zinc in the semen plasma shows protective effects [84,120].

5.2.2. Selenium

Another significant microelement is selenium, which is a component of glutathione peroxidase and thus increases the enzymatic antioxidant activity [20]. In several studies lower selenium levels in semen of infertile men were found in comparison to the healthy population. However, both the deficiency and the excess of selenium may result in fertility disorders and abnormal semen parameters [81,89-92]. Moreover, selenium has a protective effect against oxidative stress on sperm DNA, and simultaneously increases motility and sperm viability [93]. In the course of normal spermatogenesis, apart from glutathione peroxidase, selenoprotein P is the key element. In fact, the greatest amount of selenium occurs in the testis in this form.

5.2.3. Other antioxidants

In addition to selenium, vitamin C and tocopherol also present antioxidant properties by means of free radical neutralization. Therefore, it is important for the diet to be rich in vegetables and fruits, which are the main sources of these elements [88]. Furthermore, apart from its antioxidant properties, tocopherol is likely to have a protective effect against heavy metal damage [7,96]. Vegetables and fruits, especially raw green-leaved vegetables are a source of folic acid, which is important in the course of spermatogenesis, particularly in the supplementation combined with zinc [97]. According to researchers, also coenzyme Q10 may be relevant in terms of semen quality, since in its reduced form, as ubiquinol and ubisemichinone radical, it has an antioxidant effect, and is involved in all energy-dependent processes, including sperm motility. Remarkably, ubiquinone is capable of regenerating other antioxidants, such as vitamin C and vitamin E [98]. On the basis of the meta-analysis, patients receiving coenzyme Q10 showed a higher level of this substance in semen, as well as an increased concentration and better sperm motility compared to placebo [88]. However, it is possible that coenzyme Q10 delivered with the diet is not sufficient and does not result in improvement of semen quality parameters – thus, supplementation is recommended [99,100].

What is more, the supplementation of L-carnitine may also be of importance [101-103]. It has been shown to have a positive impact on sperm maturation and motility and spermatogenesis in terms of providing energy supply to sperm by transporting long-chain fatty acids to mitochondria [20,83].

Lycopene, a powerful antioxidant belonging to the carotenoid family, seems to show promising results. It is indicated that lycopene reduces lipid peroxidation and DNA damage, strengthens the immune system and increases the number and survival of sperm [120]. Zaremba et al. demonstrated a positive correlation between the consumption of lycopene and normal sperm morphology [121].

 Many beneficial properties are also attributed to N-acetyl-cysteine (NAC), which is involved in glutathione synthesis (GSH) and has the ability to capture ROS [122]. The presence of NAC in the diet of infertile men has been associated with an increased number and motility of spermatozoa, as well as an increased number of normal structure spermatozoa following 3 months of supplementation. Moreover, a decrease in sperm DNA fragmentation and an increase in protamine levels, as well as a decrease in FSH, LH and an increase in testosterone levels in blood were observed [124].

A supplementation combining many antioxidants seems particularly beneficial. All the papers focused on this issue so far have demonstrated beneficial results of multiple antioxidants on sperm parameters [19]. For instance, Gvozdjáková et al. have shown that even a 3-month supplementation with carnitine, ubiquinol and vitamins E and C has a positive effect on sperm density and motility. Additionally, the percentage of abnormal spermatozoa has also decreased [98]. It is worth emphasizing that the consumption of products with high antioxidant potential may also minimize the adverse effects of trans fats on sperm quality [45].

The most frequently used antioxidants, both in monotherapy and combined supplementation, include vitamin E and C, carnitine, coenzyme Q10, zinc, selenium, folic acid and N-acetylcysteine [88].

5.2.4. Omega-3 fatty acids

Omega-3 fatty acids, which are precursors to eicosanoids, are also known to have anti-inflammatory and antioxidant properties. Compared to other body tissues and cells, testis and spermatozoa have a higher concentration of polyunsaturated fatty acids, and effective fertilization depends on the lipid composition of the sperm membrane [7,104]. It has been demonstrated that they positively affect the concentration, number and morphology of sperm and have the ability to modify the composition of the cell membrane by building into it, thus, supporting its functioning [6,9,31,105]. It is also indicated that eicosapentaenoic acid (EPA) and/or docosahexaenoic acid (DHA) supplementation with fatty acids significantly increases sperm motility and DHA concentration in semen [106].

A meta-analysis of 16 randomized controlled trials showed a positive relationship between omega-3 supplementation and semen quality parameters in infertile men. Moreover, pro-healthy dietary models containing fish and seafood were also associated with better sperm quality in observational studies [107].

In addition, the consumption of 75g of walnuts per day over a 12 week period was associated with a longer lifespan, motility and sperm morphology. Interestingly, according to another study, the addition of 60g of nut mixture to the Western diet, apart from improving the abovementioned parameters, also resulted in an increase in sperm count [110,111].

5.3. Magnesium, calcium, copper, manganese

It is also recommended to provide an adequate supply of magnesium and calcium. The former  constitutes a key element in the course of spermatogenesis and sperm motility, also in the female reproductive tract. Furthermore, calcium affects the motility, hyperactivation and capitulation of sperm and ultimately the acrosome reaction leading to sperm penetration into the oocyte [81]. In addition, copper is also necessary for the proper functioning of sperm, and manganese affects the motility of sperm and the fertilization process [125-128]. Nevertheless, both manganese and copper in excessive amounts show an adverse effect on sperm [129,130]. The knowledge about the impact of selected trace elements and vitamins on male semen is summarized in Figure 3.

Figure 3. Selected components of male semen and their role [81].

5.4. Fibre

On the basis of the research studies, the role of fibre should also be emphasized. It is an essential element of the diet with regard to fertility due to its potential binding of non-conjugated oestrogens, which is directly associated with lower levels of oestrogen in plasma. In fact, its appropriate levels are necessary to maintain proper reproductive functions [6].

Comment: The title " 2. Poor nutritional model of male fertility" does not accurately state the topic. The syntax is improper here, Poor refers to the quality of the model as used. Please use another title. 

Reply: Thank you for that comment. According to your suggestion we  have changed title as below:

“Nutritional model increasing the risk of male infertility”

The utility of Figure 1 is not clear. In the manuscript it says "Characteristics of a diet negatively  affecting fertility and its proposed modifications is presented in Figure 1." The figure does not demonstrate clearly at all how modifications are proposed. This figure should clearly show which foods are negatively impacting fertility.

Reply: Thank you for that comment. We have made changes to the figure so it is more readable and understandable. You can see it below.

Comment: In table 1 it would be useful to make a column on how each diet modification has been shown to actually improve fertility. 

Reply: Thank you for that valuable remark. According to your suggestion, we have expanded section with comments and remarks. You can see it below:

Table 1. Characteristics of a diet beneficial for fertility

Dietary component / items

Active substances

Comments / Remarks

Oily sea fish

PUFA, omega-3

Fat-soluble vitamins - A, D, E, K

Fish and seafood represent the main sources of DHA and EPA in the diet, therefore their incorporation in the diet may be associated with the improvement of semen quality [44]. Fish are often contaminated with mercury and other neurotoxic substances [44].

Vegetables and fruit

Antioxidants, folic acid, fibre, minerals

Vegetables and fruits provide the basis for pro-healthy nutrition models, associated with the improvement of semen quality and fertility [6,7,20,25,84]. It is worth choosing raw vegetables and fruits.

Research suggests that pesticide residues may modify the beneficial effect of fruit and vegetable consumption on the quality of semen [28].

Nuts, seeds

EFAs, fibre, tocopherols, phytosterols, polyphenols, minerals

It is important to choose nuts and unroasted and unsalted seeds. The use of nuts in the diet may have a beneficial effect on the quality of sperm[110,111].

Whole-grain products

Fibre, zinc, magnesium

It is recommended to limit the consumption of refined flour products, and choose whole-grain products, rich in fibre [6,20].

Lean dairy

Calcium, a wholesome protein

It is beneficial to choose low-fat dairy products, due to a lower saturated fat content [20].

Olive oil, rapeseed oil

PUFA, alpha-linolenic acid, vitamin E, polyphenols

It is advisable to substitute saturated fats with vegetable oils containing unsaturated acid residues [6,7].

Comment: The authors discuss antioxidant supplements and promote them in a bright light as possible therapeutic options. They should fully cover this topic and discuss the evidence and actual efficacy that has been shown behind these. There are several studies in the literature evaluating the evidence behind popular supplement products. 

Reply: Thank you for that suggestion. We added a summary of knowledge about the use of antioxidants by infertile men. We are aware of the need for more extensive research and we emphasized this in the paragraph.You can read it below.

5.2. Antioxidants for male infertility – what is the evidence?

Although oxidative stress tends to be the primary factor underlying male infertility, it should be stressed that studies on the efficiency of antioxidant therapy are still contradictory. It seems that oral antioxidant supplementation improves the parameters evaluating semen quality and is associated with less DNA damage. However, no reliable studies evaluating the impact of antioxidant supplementation on positive pregnancy outcomes and live birth rate are available [116]. On the basis of 7 randomized studies, involving the live birth rate following antioxidant supplementation, it was demonstrated that such supplementation in infertile men can improve the live birth rate. However, the authors of the review emphasize that the studies are ambiguous and subject to high risk of error [117]. Tus, the need for more good quality research on efficacy, safety and the recommended doses of antioxidants was also stressed in other research studies [117-119].

We also added new references:

  1. Martin-Hidalgo,D.; Bragado, M.J.; Batista, A. R.; Oliveira, P.F.; Alves, M.G. Antioxidants and Male Fertility: from Molecular Studies to Clinical Evidence. Antioxidants 2019, 8, 89.

  1. Smits, R.M.; Mackenzie-Proctor, R.; Yazdani, A.; Stankiewicz, M.T.; Jordan, V.; Showell, M.G. Antioxidants for male subfertility. Cochrane Database of Systematic Reviews 2019.

  1. Barratt, C.L.R.; Björndahl, L.; De Jonge, C.J.; Lamb, D.J.; Osorio Martini, F.; McLachlan, R.; Oates, R.D.; van der Poel, S.; St John, B.; Sigman, M.; et al. The diagnosis of male infertility: an analysis of the evidence to support the development of global WHO guidance—challenges and future research opportunities. Human Reproduction Update 2017, 23, 660–680.

  1. Cardoso, J.P.; Cocuzza, M.; Elterman, D. Optimizing male fertility: oxidative stress and the use of antioxidants. World J Urol 2019, 37, 1029–1034.

We have also added a section on NAC and lycopene, which we did not mention at work, but are powerful antioxidants that may be relevant in the context of fertility.

Lycopene, a powerful antioxidant belonging to the carotenoid family, seems to show promising results. It is indicated that lycopene reduces lipid peroxidation and DNA damage, strengthens the immune system and increases the number and survival of sperm [121]. Zaremba et al. demonstrated a positive correlation between the consumption of lycopene and normal sperm morphology [122].

 Many beneficial properties are also attributed to N-acetyl-cysteine (NAC), which is involved in glutathione synthesis (GSH) and has the ability to capture ROS [123]. The presence of NAC in the diet of infertile men has been associated with an increased number and motility of spermatozoa, as well as an increased number of normal structure spermatozoa following 3 months of supplementation. Moreover, a decrease in sperm DNA fragmentation and an increase in protamine levels, as well as a decrease in FSH, LH and an increase in testosterone levels in blood were observed [124].

We added new references:

  1. Agarwal, A.; Durairajanayagam, D.; Ong, C.; Prashast, P. Lycopene and male infertility. Asian J Androl 2014, 16, 420.
  2. Zareba, P.; Colaci, D.S.; Afeiche, M.; Gaskins, A.J.; Jørgensen, N.; Mendiola, J.; Swan, S.H.; Chavarro, J.E. Semen quality in relation to antioxidant intake in a healthy male population. Fertility and Sterility 2013, 100, 1572–1579.
  3. Aldini, G.; Altomare, A.; Baron, G.; Vistoli, G.; Carini, M.; Borsani, L.; Sergio, F. N-Acetylcysteine as an antioxidant and disulphide breaking agent: the reasons why. Free Radical Research 2018, 52, 751–762.
  4. Jannatifar, R.; Parivar, K.; Roodbari, N.H.; Nasr-Esfahani, M.H. Effects of N-acetyl-cysteine supplementation on sperm quality, chromatin integrity and level of oxidative stress in infertile men. Reprod Biol Endocrinol 2019, 17, 24.

Figures in PDF format

Round 2

Reviewer 1 Report

The revision made improve the quality of the narrative review. I have not more comments

Reviewer 3 Report

No further changes suggested. Authors made significant improvements